# TopInG: Topologically Interpretable Graph Learning via Persistent Rationale Filtration

## Abstract

Graph Neural Networks (GNNs) have shown remarkable performance in various scientific domains, but their lack of interpretability limits their applicability in critical decision-making processes. Recently, intrinsic interpretable GNNs have been studied to provide insights into model predictions by identifying rationale substructures in graphs. However, existing methods face challenges when the underlying rationale subgraphs are complicated and variable. To address this challenge, we propose TopInG, a novel topological framework to interpretable GNNs that leverages persistent homology to identify persistent rationale subgraphs. Our method introduces a rationale filtration learning technique that models the generating procedure of rationale subgraphs, and enforces the persistence of topological gap between rationale subgraphs and complement random graphs by a novel self-adjusted topological constraint, topological discrepancy. We show that our topological discrepancy is a lower bound of a Wasserstein distance on graph distributions with Gromov-Hausdorff metric. We provide theoretical guarantees showing that our loss is uniquely optimized by the ground truth under certain conditions. Through extensive experiments on varaious synthetic and real datasets, we demonstrate that TopInG effectively addresses key challenges in interpretable GNNs including handling variiform rationale subgraphs, balancing performance with interpretability, and avoiding spurious correlations. Experimental results show that our approach improves state-of-the-art methods up to $20\%+$ on both predictive accuracy and interpretation quality. Our code is available through the link: https://anonymous.4open.science/r/TopoEx-1EE2/

## 1 Introduction

Graph Neural Networks (GNNs) have emerged as powerful tools for learning graph-structured data, in various scientific domains, such as chemistry, biology, physics, and materials science, achieving remarkable success in applications of predicting molecular properties (Kamberaj, 2022; Chen et al., 2023), modeling protein-protein interactions (Görmez et al., 2021; Ravichandran et al., 2024; Li et al., 2023), analyzing phase transitions (Qu et al., 2022), characterizing material characteristics (Hu & Latypov, 2024; Sheriff et al., 2024; Gurniak et al., 2024; Xiao et al., 2024), etc. As GNNs are increasingly applied to critical scientific and decision-making tasks, there is a growing need for interpretability and explainability in these models (Zhang et al., 2024a). Scientists and practitioners often ask for not only accurate predictions, but also insights into why and how these predictions are made. This is particularly crucial in scientific applications where understanding the underlying mechanisms and causal relationships is as important as the predictions themselves.

A recent trend in GNN research focuses on enhancing interpretability by developing methods that identify and visualize the nodes, edges, subgraphs, or features most influential or causal for a given prediction. Existing approaches to GNN interpretation can be broadly categorized into two classes (Zhang et al., 2024a): post-hoc explainer methods (Ying et al., 2019a; Luo et al., 2020a; Schlichtkrull et al., 2021; Wu et al., 2023; Bui et al., 2024) and intrinsically interpretable models (Wu et al., 2022; Miao et al., 2022; Chen et al., 2024). Post-hoc Explainer methods analyze a pre-trained GNN model to generate intuitive explanations after the fact. It enjoys flexibility and can be integrated into different kinds of models. However, recent research (Miao et al., 2022) shows that post-hoc methods might provide explanations that are suboptimal or inconsistent with the model's actual decision-making processes. On the other hand, intrinsically interpretable models incorporate

interpretability directly into the model architecture and training process. The fundamental idea of intrinsical interpretability stems from the concept of graph attention (Veličković et al., 2018). As attention weights may not always correlate with actual feature importance, a Naïve application of attenion weights is not reliable for real graph data (Ying et al., 2019a; Yu et al., 2020). Moreover, the potential trade-off between interpretability and predictive performance (Du et al., 2019) may not be acceptable in real-world applications. Therefore, various methods have been developed regarding how to use attention weights for interpretation. Miao et al. (2022) proposed stochastic attention mechanism (GSAT) to use the graph information bottleneck (Wu et al., 2020; Tishby et al., 1999) as target function, employ attention weights to control the information bottleneck, and sample rationale subgraphs using Gumbel-softmax reparameterization, to achieve strong performance in both prediction and interpretation. Similarly, Chen et al. (2024) approached interpretation by searching for rationale subgraphs within the framework of subgraph multilinear extension (SubMT) and proposing a graph multilinear net (GMT) for better SubMT approximation. Wu et al. (2022) proposed Discovering Invariant Rationales (DIR), applying interventions on training distributions to obtain invariant causal rationales while filtering out spurious correlations.

Despite these advancements, existing interpretable models often assume either explicitly or implicitly that the subgraph rationales are nearly invariant across different instances within the same category of graphs, even a strong one-to-one correspondence between subgraph rationales and predictions. However, this is overly restrictive and unrealistic in many real-world scenarios, where the graph dataset and the downstream tasks might be complicated with *varriform subgraph rationales*, which can vary significantly in form, size, and topology, even among graphs belonging to the same category. For example, in molecular biology, molecules with the same bioactivity can have different functional groups responsible for that activity Patani & LaVoie (1996); Brown (2012). An aromatic ring, a sulfonamide group, or a heterocyclic compound can each be the key substructure leading to the same pharmacological effect in different molecules. Another example can be drawn from social networks. In the scenario of identifying influential users, the structural reasons for the influence vary significantly. An influential user might have high degree centrality, being directly connected to many other users, or they might act as bridge nodes connecting different communities. Our observations, supported by experiments on a synthetic dataset we created (see Figure 3 for the results and Appendix C for the dataset construction), also show that existing intrinsically interpretable models struggle with such variability. Models obtained under these assumptions may fail to accurately capture the true causal mechanisms underlying the predictions, resulting in unreliable interpretations and bad generalization performance.

To address the above challenges, we propose *Topologically Interpretable Graph Learning* (TOPING), a novel topological approach to intrinsically interpretable GNNs that leverages techniques from topological data analysis to identify stable and persistent rationale subgraphs, effectively handling the variability in subgraph structures. Our method is inspired by the concept of persistent homology, originating from algebraic topology and recently applied to data analysis and machine learning Wong & Vong (2021); Yan et al. (2021; 2022a); Zhao et al. (2020); Immonen et al. (2023); Ye et al. (2023). Persistent homology studies the dynamics of topological invariants over various scales through a filtration process, allowing us to capture all the changes and persistence of topological features in the data.

Based on this foundation, we introduce a new perspective on the rationale subgraph identification problem. We model the graph attention mechanism as an underlying graph generation process, which ideally constructs the rationale subgraph first, followed by the addition of auxiliary structures. We use persistent homology tools to capture and track the representations and life cycles of topological features during the generating process. To effectively distinguish the rationale subgraph from the complement subgraph, we optimize the parameterized generation procedure to enhance the stability of the rationale subgraph. Specifically, our goal is to amplify the topological differences between the rationale subgraph and the complement subgraph, creating a persistent gap in their topological features throughout the generation process. To achieve this goal, we propose a novel self-adjusting topological constraint, *topological discrepancy*, which measures the statistical difference between two graphs with respect to their topological structures. The topological discrepancy serves as a metric to quantify how well the rationale subgraph is preserved and distinguished from the complement subgraph during the filtration process. We also provide a tractable approximation of our topological discrepancy and provide theoretical guarantees that our models are able to achieve ground truth as the unique optimal solution under our loss function.

Our main contributions of the paper can be briefly summarized as follows:

- We introduce TOPING, a novel intrinsically interpretable GNN framework that incorporates topological data analysis to identify stable rational subgraphs via persistent rationale filtration learning. We propose a new loss function, *topological descrepency*, to measure the statistical difference between two graphs with respect to their topological structures.

- We provide a tractable approximation of our topological discrepancy and provide theoretical guarantees that our models are able to achieve ground truth as the unique optimal solution under our loss function. This establishes a solid theoretical foundation for the effectiveness of our approach.

- We empirically demonstrate that TOPING improves existing methods in both prediction and interpretation tasks on multiple benchmark datasets, up to $20\%+$ on both interpretation and prediction performance. Additionally, we created a synthetic dataset with variiform rationale subgraphs to specifically target challenges faced by previous methods. Our results show that TOPING effectively handles such variability, confirming its ability to address this critical challenge.

## 2 PRELIMINARY

### 2.1 GRAPH NEURAL NETWORKS (GNNs)

Graph neural networks are a class of neural networks designed to operate on graph-structured data. A typical message-passing GNN layer updates node representations by aggregating information from neighboring nodes:

$$h_v^{(l+1)} = \phi(h_v^{(l)}, AGG(h_u^{(l)} : u \in N(v)))$$
(1)

where $h_v^{(l)}$ is the message representation of node $v$ at layer $l$, $N(v)$ is the neighborhood of $v$, AGG is an permutation invariant aggregation function, e.g.: sum, mean, max, and $\phi$ is a non-linear activation function. Some commonly used graph neural networks architecture includes Graph Convolutional Networks (GCN) (Kipf & Welling, 2017), Graph Isomorphism Networks (GIN) (Xu et al., 2019), Graph Attention Networks (GAT) (Veličković et al., 2018).

### 2.2 INTRINSICALLY INTERPRETABLE GRAPH LEARNING

Intrinsically interpretable graph learning aims to build a model simultaneously targeting for both performance and interpretability during the training procedure. Formally, given a collection of labeled graphs $(\mathcal{G}, Y) = \{(G, y_G)\}$, assume each graph $G$ is composed with two edge disjoint subgraphs $G = G_X \sqcup G_\epsilon$ with vertex correspondence for some $G_X \in \mathcal{G}_X$ and $G_\epsilon \in \mathcal{G}_\epsilon$. $\mathcal{G}_X$ and $\mathcal{G}_\epsilon$ are two families of graphs. $\mathcal{G}_X$ is usually a small finite set. Given a graph $G$, $G_X$ is the rationale subgraph in $G$ that almost determines the label $y_G \approx h^*(G_X)$ up to some random noise, for some unknown oracle $h^* : \mathcal{G} \to [0, 1]$. $G_\epsilon$ is the noise or less relevant part of in the graph. Both $G_X$ and $G_\epsilon$ are unknown and they have to be learned from the data. The goal is to predict the label $\hat{y}_G$ for each graphs $G$ and simultaneously identify its rationale subgraphs $G_X$.

### 2.3 TOPOLOGICAL DATA ANALYSIS (TDA)

Recent year, TDA has found its applications in various areas such as machine learning, artificial intelligence, data science, neuroscience, an so on (Giunti et al., 2022). Especially in the area of graph representation learning, TDA has shown the power of enhancing popular GNNs on different tasks by augumenting potentially useful topological features represented by TDA methods Hofer et al. (2017); Dehmamy et al. (2019); Carrière et al. (2020); Horn et al. (2022). One successful tool is *persistent homology*. On graphs, the persistent homology is mainly determined by a graph filtration which is usually induced by some edge filtration function. With the spirit of machine learning, it is natural to consider learning the edge filtration function from data to search for an optimal filtration for downstream tasks. Along this approach, various models have been proposed for the graph filtration learning (Carrière et al., 2020; Horn et al., 2022; Hofer et al., 2020; Xin et al., 2023; Yan et al., 2022a; Zhao et al., 2020; Carrière & Blumberg, 2020; Zhang et al., 2024b). Just

name a few. We give a brief introduction to the basic concepts of topological data analysis (TDA) and persistent homology, which are essential for understanding our proposed method. For a more detailed introduction, we refer readers to (Edelsbrunner & Harer, 2010; Dey & Wang, 2022).

For an edge weighted graph $G = (V, E, f : E \to \mathbb{R})$, we define a graph filtration as an increasing sequence of nested subgraphs $\mathcal{F}(G) := \{G_{\leq t} \mid t \in f(E)\}$, where $G_{\leq t} = (V, E_t)$ with $E_t = \{e \in E : f(e) \leq t\}$. By tradition, set $G_{-\infty} = \emptyset$ and $G_{+\infty} = G$ to be the first and the last element in $\mathcal{F}(G)$. On such a filtration, applying $p$-homology functor (Hatcher, 2002), $H_p(\mathcal{F}(G))$ outputs a chain of homology groups (vector spaces over fields)

$$H_p(\mathcal{F}(G)) : 0 \to \cdots \to H_p(G_{\leq t_1}) \to H_p(G_{\leq t_2}) \to H_p(G_{\leq t_3}) \to \cdots \to H_P(G)$$

connected by linear maps naturally induced by inclusion maps. Such an algebraic structure is called a *persistent homology*. In this paper, we only consider $p = 0, 1$ which corresponding to connected components and cycle bases in graphs. We use the finite field $\mathbb{F}_2$ as the coefficient field for homology groups. Then, the $p$-th persistent homology group $H_p(\mathcal{F}(G))$ is a sequence of vector spaces over $\mathbb{F}_2$ with linear maps between them. Essentially, persistent homology captures the evolution of persistent topological features (e.g., connected components, cycles, voids, ...) in the graph filtration. These topological features can be summarized as a complete discrete invariant known as persistence diagram (Edelsbrunner & Harer, 2010; Carlsson et al., 2009), $\mathrm{PD}(G)$, which is a collection of points in $\mathbb{R}^2$. Each point in the persistence diagram essentially represents the lifecycle (birth, death) of a persistent topological feature. We provide a concrete example in Figure 1 to illustrate intuitive ideas behind TDA.

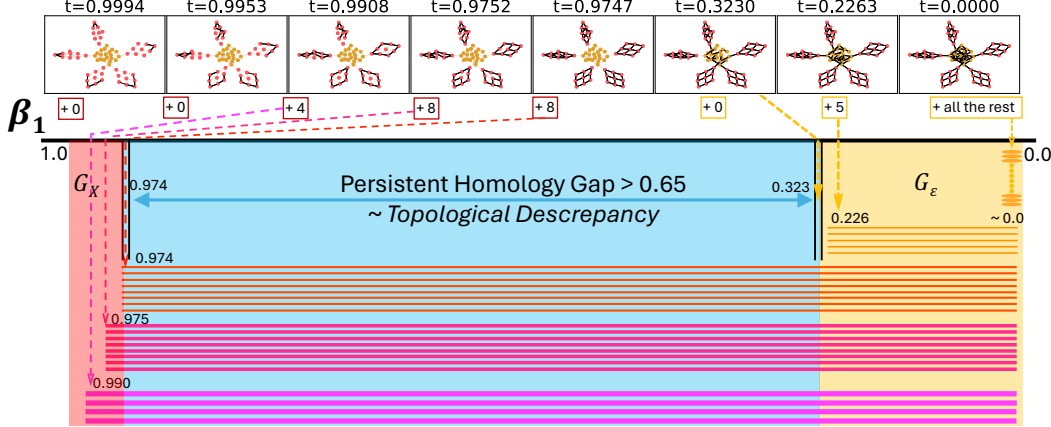

Figure 1: The top row sequence is our learned rational filtration on an example graph. Red and yellow points correspond to ground truth rationale subgraph $G_X$ and noisy subgraph $g_\epsilon$ respectively. Each snapshot is a subgraph $G_{\leq t}$ with $t$ showing on the top of the figure. We did not do any normalization on the filtration values. Observed that all edges in $G_X$ have weights $\geq 0.974$ and all edges in $G_\epsilon$ have weights $\leq 0.323$, which means the rationale filtration we learned is quite consistent with the ground truth rationale. Taking a closer look at the filtration, one can see that, the generating procedure of the rationale filtration is well-ordered and fast. There is a clear pattern of the generating procedure. However, the noisy graph is generated in a more chaotic way. Only five cycles are generated until $t$ reaches 0.226. Most of the cycles are generated until $t$ closed to 0. Now check the bottom part of the figure showing the barcode of the filtration, which is a topological summary equivalent to persistence diagram. Each horizontal bar corresponds to a topological feature. Here we only illustrate 1-st degree persistent homology, which correponds to cycle bases in graphs. The left end of each bar indicates the first time $t$ it appears in the filtration. The most important information one can get from the barcode is that, within the interval $[0.974, 0.226]$, the barcode does not change at all. That means the persistent topological structure of the graph is stable within this region. The length of the interval $(0.974 - 0.226)$ is what we called *persistent homology gap*, which is a measure of the difference of topological structures between the $G_X$ and $G_\epsilon$. Note that this gap is very closed to the gap between minimal edge weight in $G_X$ and maximal edge weight in $G_\epsilon$. In fact, they will be exactly the same if we also consider the 0-th persistent homology. Such gap will be approximated by what we proposed *topological descrepancy*, and our final target function is designed to maximize persistent homology gaps statistically over all data.

Persistence diagrams are topological invariants. Therefore, they can be viewed as graph representations that capture topological structures of input graphs. Two persistence diagrams $\mathrm{PD}(G), \mathrm{PD}(G')$

can be compared through the bottleneck distance $d_{\mathrm{B}}$ (Edelsbrunner & Harer, 2010), which is defined as the Wasserstein distance between the two persistence diagrams viewed essentially as two collections of points in the $\mathbb{R}^2$. The bottleneck distance is a (pseudo)metric that quantifies the similarity between two persistence diagrams, also similarity between two graphs with respect to their topologies.

One important property of the bottleneck distance is that it is stable under perturbations of the input data. Formally, the bottleneck distance is stable with respect to the Gromov-Hausdorff distance (Chazal et al., 2009) on graphs. Given two finite, connected, weighted (with no negative cycle) graphs $G$ and $H$, the Gromov-Hausdorff distance between $G$ and $H$ is defined as

$$d_{\mathrm{GH}}(G, H) = \frac{1}{2} \inf_{\Pi} \sup_{(u,v),\,(u',v') \in \Pi} | \, d_G(u, u') - d_H(v, v') \, |, \tag{2}$$

where $\Pi \subseteq V_G \times V_H$ is a coupling such that $\Pi_1 = V_G$ and $\Pi_2 = V_H$, and $d_G, d_H$ are shortest path distances on $G$ and $H$ respectively. The Gromov-Hausdorff distance measures the distortion between two graphs. Intuitively, for two isomorphic graphs, the Gromov-Hausdorff distance is zero. It is known that $d_{\mathrm{B}} \leq 2 d_{\mathrm{GH}}$.

## 3 METHOD OF TOPING

In the following context, for a given $G$, we denote the oracle rationale subgraph and its complement as $G_X^*$ and $G_\epsilon^*$. Use $G_X$ and $G_\epsilon$ to represent a candidate rationale subgraph predicted by our model.

In contrast to existing methods, our approach reconsiders the problem from a more global perspective through the lens of topology. If the graph classification/prediction task indeed can be captured by a rationale subgraph as a core structure in a relatively small family $\mathcal{G}_X$, then the graph $G$ can be considered as 'growing' from the core with additional auxiliary structures attached to the core. Nevertheless looking for this core rationale substructure is highly non-trivial, as this requires maintaining consistency across the growing procedure (i.e., not losing edges in the middle) and identifying common subgraphs across many instances in the data input. We propose to learn a *filtration function* that captures the importance of edges in the graph *generating* process, allowing us to identify *stable and persistent* substructures that are most relevant for predictions. This approach aims to leverage the power of topological data analysis to improve the interpretability and generalization of GNNs while maintaining high predictive performance.

Based on our assumption, we consider a generating process of $G = (V, E)$ by first generating the most important part which corresponds to the candidate rationale subgraph $G_X$, and then combined with some noisy graph $G_\epsilon$ as complement to get the final graph $G$. Following this idea, for a given graph $G$, we consider a filtration on the graph $\mathcal{F}(G)$ which is a sequence of step-by-step generating process of $G$ based on an ordering of the edges in $G$. More precisely, we construct an ordering on edges $(e_1, e_2, \cdots, e_{|E|})$ and induced graph filtration $\mathcal{F}(G) = \{G_0, G_1, \ldots, G_{|E|}\}$, where $\forall i \in [|E|]$, $G_i = G_{i-1} \cup e_i$ with $G_0$ initialized to be the empty graph and $G_{|E|} = G$. Intuitively, we hope such ordering can capture the importance of the edges in $G$. We use a filtration function $f : E \to [0, 1]$ to represents the importance of each edge in $G$ and the order of edges is following $1 - f(e)$. It is natural to assume that the more important the edge is, the earlier it appears in the ordering (and those pairs $(u, v) \notin E$ are out of the generating process). Following such idea, we also require this ordering to be consistent with the importance of $G_X$ and $G_\epsilon$. That is to say, $f(e \in G_X) > f(e' \in G_\epsilon)$ . In our model, we will learn a *filtration functional* $f_\phi : G \to [0, 1]^{|E|}$ to construct for each graph a function $f_\phi^G : E \to [0, 1]$ mapping edges to their importance score. For concise of notations, we might omit the upper and lower indices for $f = f_\phi^G$ and $\mathcal{F}(G) = \mathcal{F}_\phi(G)$ if they are clear in the context. We denote the subfiltration $\mathcal{F}(G_{\leq t}) \subseteq \mathcal{F}(G)$ to be the filtration consisting of the subgraphs in $\mathcal{F}_\phi(G)$ whose edges' filtration values are all below or equal $t$. Symmetrically, let $\mathcal{F}(G_{\geq t})$ to be the filtration consisting of the subgraphs whose edges' filtration values are all above $t$.

Before we talk about the construction of our model, let us first discuss what ideal properties we are looking for in our filtration function $f_\phi$. Let $\mathcal{F}(\mathcal{G}) = \{\mathcal{F}(G) : G \in \mathcal{G}\}$ be the collection of all graph filtrations determined by $f_\phi$. For a given $t \in \mathbb{R}$, denote $\mathcal{F}(\mathcal{G}_{\leq t}) := \{\mathcal{F}(G_{\leq t}) : G \in \mathcal{G}\}$ and $\mathcal{F}(\mathcal{G}_{\geq t}) := \{\mathcal{F}(G_{\geq t}) : G \in \mathcal{G}\}$. We consider the following property:

**Topological Discrepancy**: There exists a global threshold $t$ such that, the distributions of $\mathbb{P}(\mathcal{T} \circ \mathcal{F}(G_{\leq t}))$ and $\mathbb{P}(\mathcal{T} \circ \mathcal{F}(G_{\geq t}))$ are discrepant with respect to some topological invariants $\mathcal{T}$.

**Remark 3.1.** *The underlying idea of this property is that, if we track the generating process of the rationale subgraph $G_X$ and the noise subgraph $G_\epsilon$, we hope to see in general two very different evolutionary paths on the topological structures during the process. The methods based on our persistent rationale filtration framework should be able to capture such difference.*

$$\mathcal{L}(\phi) = \mathbb{E}_G\left[\mathcal{R}(h_\phi \sigma f_\phi(G))\right] - \alpha d_{\text{topo}}(\mathbb{P}(\mathcal{T} \circ \mathcal{F}_\phi(G_{\leq t})), \mathbb{P}(\mathcal{T} \circ \mathcal{F}_\phi(G_{\geq t}))) \tag{3}$$

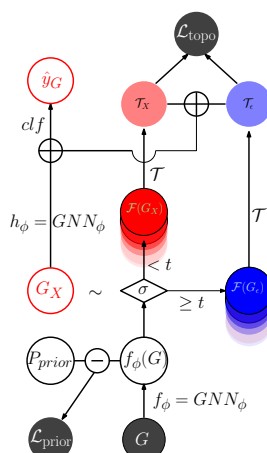

Formally, we consider persistence diagrams as our topological invariants $\mathcal{T}$. We define the *topological discrepancy* $d_{\text{topo}}$ between $P = \mathbb{P}(\mathcal{T} \circ \mathcal{F}(G_{\leq t}))$ and $Q = \mathbb{P}(\mathcal{T} \circ \mathcal{F}(G_{\geq t}))$ as follows:

$$d_{\text{topo}}(P, Q) \triangleq \inf_{\pi \in \Pi(P,Q)} \mathbb{E}_{(p,q)\sim\pi}[d_{\text{B}}(p, q)] \tag{4}$$

Essentially, $d_{\text{topo}}$ is the 1-Wasserstein distance between the distributions of induced persistence diagrams of subfiltrations $\mathcal{T} \circ \mathcal{F}(G_{\leq t})$ and $\mathcal{T} \circ \mathcal{F}(G_{\geq t})$ with metric $d_{\text{B}}$.

Now we are ready to design a high-level model together with a loss function approximating an $f_\phi^*$ based on our topological discrepancy property. See Figure 2 as an illustration. We use a GNN model to learn the filtration function $f_\phi$. After that, we apply some extraction function $\sigma$ to separate graph $G$ into two subgraph $G_X \sqcup G_\epsilon$. For simplicity, one can just consider $\sigma$ to be a hard cut with threshold value $t = 0.5$. Based on the extracted $G_X = G_{<0.5}$, we apply a GNN model $h_\phi$ followed by a classifier to predict the label $y_G$. Here we use the same GNN model with shared parameters from $f_\phi$. The classifier is some MLP whose parameters are omitted in the loss function for simplicity. The loss function $\mathcal{R}$ is the standard cross-entropy loss between the predicted label and the ground truth.

Figure 2: TOPING uses a GNN to learn a filtration functional $f_\phi$. It extracts graph filtrations of $G_X$ and $G_\epsilon$ and use them to compute topological features. $G_X$ is sampled from $f_\phi$. Sampled $G_X$ passes the same GNN with shared parameter of $f_\phi$ to get a graph representation. Concatenated with topological features (which are naturally global feature of the graph), model get final representation before the classifier.

### 3.1 SELF-ADJUSTED TOPOLOGICAL CONSTRAINT

In this subsection, we will discuss the construction and properties of our topological features in details. For briefness, we denote the distribution of persistence diagrams $\mathcal{P}(G_X) := \mathbb{P}(\mathcal{T} \circ \mathcal{F}(G_{<t}))$ and $\mathcal{P}(G_\epsilon) := \mathbb{P}(\mathcal{T} \circ \mathcal{F}(G_{\geq t}))$ respectively. In summary, we will show an upper and lower bound of our $d_{\text{topo}}$ as follows:

**Theorem 3.2.** *Given a finite collection of 1-Lipschitz continuous functions, $\Psi = \{\psi_1, \psi_2, \cdots\}$, on the space of persistence diagrams, $d_{topo}$ have an upper and lower bound as follows:*

$$\max_{\psi \in \Psi} |\mathbb{E}_{P \sim \mathcal{P}(G_X)}[\psi(P)] - \mathbb{E}_{Q \sim \mathcal{P}(G_\epsilon)}[\psi(Q)]| \leq d_{topo}(\mathcal{P}(G_X), \mathcal{P}(G_\epsilon)) \leq 2d_{wass}(\mathbb{P}(G_X), \mathbb{P}(G_\epsilon))$$

*Proof.* The upper bound is from the $d_{\text{GH}}$-stability property of bottleneck distance $d_{\text{B}}$ on persistent diagrams. The lower bound is from the Kantorovich duality of Wasserstein distance (Villani, 2009):

$$d_{\text{wass}}(P, Q) = \sup_{\|\psi\|_{\text{Lip}} \leq 1} |\mathbb{E}_{p \sim P}[\psi(p)] - \mathbb{E}_{q \sim Q}[\psi(q)]|$$

$\square$

Essentially, the upper bound says that topological discrepancy has discriminative power up to the Wasserstein distance between the marginal distributions of $G_X$ and $G_\epsilon$. For the lower bound, we will use it to derive a tractable approximation of $d_{\text{topo}}$ in practice. We apply a family of learnable

vectorization functions introduced by Hofer et al. (2019) to represent persistence diagrams as some $k$-dimensional vectors. These functions are Lipschitz continuous. More details about the construction can be found in the appendix B. Based on above, we have a tractable lower bound approximation of our $d_{\text{topo}}$.

**Remark 3.3.** *In practice the expections are approximated by the empirical means. The maximum can be picked out by a softmax attention during training. Here we instead apply a 2-head attention mechanism to select the top-2 maximums and add them up. We use $k = 8$ in our experiments. Intuitively, the vectorization function together with multi-head attentions not only provide a lower bound approximation of $d_{topo}$ for the sake of efficient computation, but also a **self-adjusted** focus on data dependent topological features. Essentially, it will help the model to learn the most relevant topological features for the downstream tasks. In practice, we found that it not only makes the training procedure more stable, but also leads to a better performance. All the topological representations we used are Lipschitz continuous, hence differentiable almost everywhere. We use the code in Zhang et al. (2024b) to compute our topological representations and gradients.*

Finally, we give the following theorem to show when our model is guaranteed to be optimized by the ground truth. The proof is a bit technical. We provide it in the appendix B.

**Theorem 3.4.** *Assume $\forall G$, $|E_X| < |E_\epsilon|$, and $G_X^*$ is minimal with respect to $y_G$ in the sense that any subgraph $G_X \subset G_X^*$ losses some information of label, then $d_{topo}$ is uniquely optimized by $f_\phi^*(e) = 1\{e \in G_X^*\}$.*

**Remark 3.5.** *Note that our guarantee does not depend on any stability or invariance assumptions on $G_X$, therefore, it will not be affected by variiform rationale subgraphs in theory.*

### 3.2 PRIOR REGULARIZATION

Despite the theoritical guarantee we provide, in practice, more powerful model does not neccessarily imply better performance in general. We found sometimes our model can still overfit. We add a prior regularization term on our edge filtration $f_\phi$. It significantly helps stabilize the training procedure.

$$\mathcal{L}(\phi) + \beta\mathcal{L}_{prior}(f_\phi(G), \mathbb{P}_{prior}) \tag{5}$$

We set for a prior marginal distribution on edge filtration $\mathbb{P}_{prior} = 0.5(\mathcal{N}(\mu_1, r_1) + \mathcal{N}(\mu_2, r_2))$ with $\mu_1, \mu_2 = 0.25, 0.75$ and $r_1, r_2$ being learnable parameters initialized with $0.25$. Then the prior regularization term $\mathcal{L}_{prior}$ is calculated as:

$$\mathcal{L}_{prior}(f_\phi(G), \mathbb{P}_{prior}) = D_{\text{KL}}[f_\phi(G)\|\mathbb{P}_{prior}] + \gamma(r_1^{-2} + r_2^{-2}) \tag{6}$$

$$= -\sum_{e \in G_E} \log(\mathbb{P}_{prior}(f_\phi(G)_e)) + \gamma(r_1^{-2} + r_2^{-2}) \tag{7}$$

The term $\gamma(r_1^{-2} + r_2^{-2})$ is added to the KL divergence to prevent the model from collapsing to a single mode. In practice, we found that Gumbel-Softmax reparameterization trick (Jang et al., 2017) used in (Miao et al., 2022) sometimes also helps stabilize the training procedure.

**Remark 3.6.** *Although in this section, we only talk about edge filtrations, our methods can be extended to filtrations on nodes, edges, and higher order simplices (faces, tetrahedrons, etc.). In fact, in our experiments we just use node filtration and extend it to edges by setting $f(u, v) = \min(f(u), f(v))$ or $\max(f(u), f(v))$. This is called upper-star or lower-star filtration in TDA. Obviously it contains less information in general since node filtrations can only represents $O(|V|)$ much "information" but edge filtrations can represents up to $O(|E|) = O(|V|^2)$ "information". We do this mainly because it speed up the computation of persistent homology based on our currently used tool package (Zhang et al., 2024b). From the proof of our Theorem 3.4 we know that using the node filtration is in fact enough to guarantee the optimization solution. The performance of our experimental results is also good enough. But of course, in general, using both node and edge filtrations would give the model more power.*

### 3.3 RELATED WORK

Two works are most related to ours: DIR (Wu et al., 2022) and GSAT (Miao et al., 2022).

Compared to DIR, our model also considers the distribution of the complement graph, but in a "soft way", which is more efficient since we do not store those graphs exactly. Intuitively, our methods

can be viewed as storing a learnable distribution of topological summary of the complement graphs. Also, in practice we do not use a hard threshold to filter the graphs. What we do is computing the persistent homology along ascending ordering and descending ordering separately, to mimic a hard cut for some threshold $t$. Since our TDA method is robust enough, in practice it works good.

Compared to GSAT, our loss function can also be viewed as a variational lower bound of the GIB loss. However, we use a totally different prior distribution of the rationales $G_X$, and get rid of the hyperparameter $r$ used in GSAT to specify the mean values of edge attentions. Instead, our topological loss can be viewed as a self-adjusted cut to separate $G_X$ from $G$. In practice, we observe that the attention learned by GSAT can collapse to the constant value $r$ if it is not tuned carefully, which is also mentioned in Chen et al. (2024). But our method does not have this issue. We consider that such an issue might be caused by the unimodality of the prior distribution used in GSAT. However, our prior is bimodal, which is essentially doing an unsupervised clustering over two Gaussians, like $k$-means. In practice, we find the position of the two centers of the prior distribution does not matter too much, as long as they do not collapse into one. Therefore, we just fix them to be $0.25$ and $0.75$, with a penalty term to prevent component collapse.

## 4 EXPERIMENTS

We evaluate our proposed method in terms of both interpretability and predictive performance on the seven most commonly used datasets. Our approach, TOPING, demonstrates significant advantages over state-of-the-art post-hoc interpretation methods as well as inherently interpretable models across almost all datasets. We will provide a brief introduction to the datasets, baselines, and experiment setups, and leave more details in the Appendix C.

### 4.1 EXPERIMENTAL SETTINGS

**Datasets.** We consider eight datasets commonly referenced in the graph explainability literature and classify them into *Single Motif*, *Multiple Motif* and *Real Dataset*. For *Single Motif*, we consider BA-2Motifs (Luo et al., 2020b), BA-HouseGrid (Amara et al., 2023), SPmotif0.5 and SPmotif0.9 (Wu et al., 2022). These datasets contain graphs with a single type of motif or structural pattern repeated throughout. For *Multiple Motif*, we consider BA-HouseAndGrid, BA-HouseOrGrid (Bui et al., 2024), and BA-HouseOrGrid-nRnd. The last one is a synthetic dataset we create for verifying the variiform rationale challenge for existing intrinsic methods(see Appendix C). These datasets involve graphs with multiple types of motifs, thereby increasing the complexity and providing a more challenging scenario for explanation methods. For *Real Dataset*, we include Mutag (Luo et al., 2020b) and Benzene (Sanchez-Lengeling et al., 2020) for interpretation.

**Baselines.** We evaluate the interpretability of several methods by differentiating between post-hoc and inherently interpretable approaches. The post-hoc methods we compare include GNNExplainer (Ying et al., 2019a), PGExplainer (Luo et al., 2020b), MatchExplainer (Wu et al., 2023), and Mage (Bui et al., 2024). Additionally, we consider the inherently interpretable methods DIR (Wu et al., 2022), GSAT (Miao et al., 2022), and GMT-Lin (Chen et al., 2024), known for their state-of-the-art interpretation capabilities and generalization performance.

**Setup.** Since we focus on graph classification tasks, GIN (Xu et al., 2018) is used as the backbone model for baselines. Furthermore, in order to support more general filtrations beyond nodes and edges, i.e, data supported on topological domains such as simplicial complexes (Bodnar et al., 2021b), cell complexes (Bodnar et al., 2021a), and even hypergraphs (Chien et al., 2022). We first apply CINPP (Giusti et al., 2023) as our backbone to test the wide applicability of TOPING.

**Metrics and evaluation.** For interpretation evaluation, we report explanation ROC AUC following (Ying et al., 2019b; Luo et al., 2020b). For prediction performance, we report classification accuracy for real datasets and SPmotif (Wu et al., 2022) for generalization performance. All the results are averaged over 5 times tests with different random seeds. All methods adopt the same graph encoder and optimization protocol to ensure fair comparisons. We set the hyperparameters according to the recommendations of previous work.

## 4.2 RESULT COMPARISON AND ANALYSIS

**Variiform Rationale Challenge.** As shown in Figure 3, the interprebility of two SOTA existing intrinsic methods decrease drastically when the nubmer of rationale graphs increase. Our method's performance is much better and stable among varriform rationale dataset.

**Interpretation performance.** As shown in Table 1, compared to the most post-hoc based methods(in the first row), and latest intrinsic interpretable models(in the second row), TOPING has shown significant improvement across almost all datasets. Especially on the Spurious-Motif datasets, which are challenging due to spurious correlations in the training data, we achieve nearly a 20% improvement over the previous best approach. On the challenging Multiple Motif and Benzene datasets, TOPING even achieves the best performance.

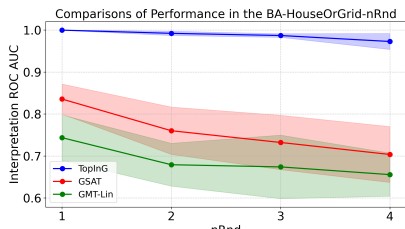

**Prediction performance.** We compare the results of all intrinsic interpretable models training from scratch. Table 2 shows the prediction accuracy on *Real Dataset* and *Spurious Motif*. TOPING significantly outperforms other baseline models on the Spurious-Motif datasets, which exhibit varying degrees of spurious correlations. This supports our claim that the model can more effectively focus on classifying the optimal stable subgraph through persistent rationale filtration learning.

Figure 3: In BA-HouseOrGrid-nRnd dataset, as $n$ grows, the number of rationale subgraphs increases. Existing intrinsically interpretable methods face significant difficulties in learning these interpretable subgraphs.

Table 1: Interpretation Performance (AUC) on test datasets. The shadowed entries are the results with mean-1*std larger than the mean of the corresponding best baselines.

| Method | SingleMotif | | | | MultipleMotif | | RealDataset | |
|---|---|---|---|---|---|---|---|---|
| | **BA-2Motifs** | **BA-HouseGrid** | **SPmotif0.5** | **SPMotif0.9** | **BA-HouseAndGrid** | **BA-HouseOrGrid** | **Mutag** | **Benzene** |
| GNNEXPLAINER | $67.35 \pm 3.29$ | $50.73 \pm 0.34$ | $62.62 \pm 1.35$ | $58.85 \pm 1.93$ | $53.04 \pm 0.38$ | $53.21 \pm 0.36$ | $61.98 \pm 5.45$ | $48.72 \pm 0.14$ |
| PGEXPLAINER | $84.59 \pm 9.09$ | $50.92 \pm 1.51$ | $69.54 \pm 5.64$ | $72.34 \pm 2.91$ | $10.36 \pm 4.37$ | $3.14 \pm 0.01$ | $60.91 \pm 17.10$ | $4.26 \pm 0.36$ |
| MATCHEXPLAINER | $86.06 \pm 28.37$ | $64.32 \pm 2.32$ | $57.29 \pm 14.35$ | $47.29 \pm 13.39$ | $81.67 \pm 0.48$ | $79.87 \pm 1.61$ | $91.04 \pm 6.59$ | $55.65 \pm 1.16$ |
| MAGE | $79.81 \pm 2.27$ | $82.69 \pm 4.78$ | $76.63 \pm 0.95$ | $74.38 \pm 0.64$ | $99.95 \pm 0.06$ | $99.93 \pm 0.07$ | $99.57 \pm 0.47$ | $96.03 \pm 0.63$ |
| DIR | $82.78 \pm 10.97$ | $65.50 \pm 15.31$ | $78.15 \pm 1.32$ | $49.08 \pm 3.66$ | $64.96 \pm 14.31$ | $59.71 \pm 21.56$ | $64.44 \pm 28.81$ | $54.08 \pm 13.75$ |
| GSAT | $98.85 \pm 0.47$ | $98.58 \pm 0.59$ | $74.49 \pm 4.46$ | $65.25 \pm 4.42$ | $92.92 \pm 2.03$ | $77.52 \pm 3.71$ | $99.38 \pm 0.25$ | $91.57 \pm 1.48$ |
| GMT-LIN | $97.72 \pm 0.59$ | $85.68 \pm 2.79$ | $76.26 \pm 5.07$ | $69.08 \pm 10.14$ | $76.12 \pm 7.47$ | $74.36 \pm 5.41$ | $\mathbf{99.87 \pm 0.09}$ | $83.90 \pm 6.07$ |
| TOPING | $\mathbf{100.00 \pm 0.00}$ | $\mathbf{99.87 \pm 0.13}$ | $\mathbf{95.08 \pm 0.82}$ | $\mathbf{90.82 \pm 4.95}$ | $\mathbf{100.00 \pm 0.00}$ | $\mathbf{100.00 \pm 0.00}$ | $96.38 \pm 2.56$ | $\mathbf{100.00 \pm 0.00}$ |

Table 2: Prediction Performance (Acc) on test datasets. The shadowed entries are the results with mean-1*std larger than the mean of the corresponding best baselines.

| | RealDataset | | SpuriousMotif | | |
|---|---|---|---|---|---|
| | **Mutag** | **Benzene** | **b=0.5** | **b=0.7** | **b=0.9** |
| DIR | $68.72 \pm 2.51$ | $50.67 \pm 0.93$ | $45.49 \pm 3.81$ | $41.13 \pm 2.62$ | $37.61 \pm 2.02$ |
| GSAT | $\mathbf{98.28 \pm 0.78}$ | $\mathbf{100.00 \pm 0.00}$ | $47.45 \pm 5.87$ | $43.57 \pm 2.43$ | $45.39 \pm 5.02$ |
| GMT-LIN | $91.20 \pm 2.75$ | $\mathbf{100.00 \pm 0.00}$ | $51.16 \pm 3.51$ | $53.11 \pm 4.12$ | $47.60 \pm 2.06$ |
| TOPING | $92.92 \pm 7.02$ | $\mathbf{100.00 \pm 0.00}$ | $\mathbf{79.30 \pm 3.92}$ | $\mathbf{75.46 \pm 7.62}$ | $\mathbf{65.64 \pm 4.98}$ |

**Ablation Studies.** In addition to the interpretability and generalizability analysis, we also conduct further ablation studies to gain a deeper understanding of the results. Table 3 illustrates the usefulness of topological regularizer and the prior Guassion regularizer. Topological constraint is essential for finding more complex subgraphs, but it struggles with classification performance. Gaussian prior distribution can successfully partition a graph, but it lacks the ability to accurately identify interpretable subgraphs. We also examine the hyperparamter sensitivity of them in BA-HouseAndGrid dataset. As is shown in Fig. 4, TOPING maintains stronger robustness against prior regularization choices compared to the topological constraint. However, using too large or too small topological regularizer weights can negatively affect both interpretation performance and prediction accuracy.

Table 3: Ablation studies.

| Method | BA-2Motifs | | BA-HouseGrid | |
| --- | --- | --- | --- | --- |
| | ACC | AUC | ACC | AUC |
| TOPING w/o $d_{\text{topo}}$ | $100.00 \pm 0.00$ | $97.90 \pm 1.24$ | $89.24 \pm 5.40$ | $92.17 \pm 6.43$ |
| TOPING w/o $\mathcal{L}_{prior}$ | $53.49 \pm 4.03$ | $93.20 \pm 4.61$ | $52.10 \pm 1.72$ | $98.76 \pm 1.53$ |
| TOPING | $100.00 \pm 0.00$ | $100.00 \pm 0.00$ | $100.00 \pm 0.00$ | $99.87 \pm 0.13$ |

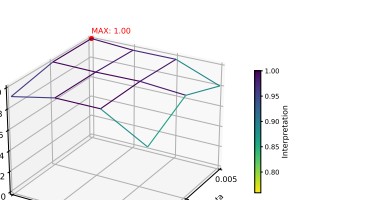 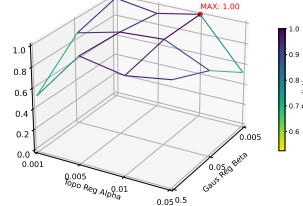

(a) Sensitivity of Interpretation(AUC)          (b) Sensitivity of Prediction(ACC)

Figure 4: A sensitivity study on BA-HouseAndGrid shows results with the topological constraint coefficient varied from [0.001, 0.005, 0.01, 0.05] and the coefficient of prior regularization term from [0.005, 0.05, 0.5].

## 5 CONCLUSION AND FUTURE WORK

In this work, we reconsider the intrinsically interpretable graph learning problem via learning a persistent rationale filtration. We propose our novel TOPING model that leverages the persistent homology to represent topological features of graphs. Based on that, we propose a novel self-adjusted topological constraint, topological discrepancy, to measure the statistical topological difference between two graph distributions. We provide a theoretical guarantee that our target function can be uniquely optimized by ground truth under certain conditions. We empirically show that our model can handle a newly targeted challenge on one simple synthetic dataset. From experiments, we also see that our model can solve other challenges including balancing performance of interpretability and prediction and avoiding spurious correlations.

### 5.1 LIMITATION

One limitation of our model is the computational cost. Currently the bottleneck is limited by the computation of the topological invariants. The main technique issue is that there is no efficient GPU implementation of the core algorithm to compute the persistent homology. The data transfer between GPU memory and CPU memory takes much I/O cost. Maybe some system-level optimization based on the CUDA framework can help. Some attempts have been made to use GPU to accelerate the computation of persistent homology (Zhang et al., 2020), but the performance is still not satisfactory enough. Another possible solution is to use some approximation algorithms to compute the topological invariants. For example, some efficient sparsification methods (Dey et al., 2019), or pretained NNs for computing persistent homology (Yan et al., 2022b). We leave these problems for the future.

### 5.2 FUTURE WORK

Another potential extension is to use multi-parameter filtration. Our current model is based on a linear graph filtration, which is based on the assumption that the importance of the edges in a graph generating procedure is a 1-dimensional scalar. But if multi-dimensional vector can represents a more sophisticated importance relations among all edges, which should in theory enrich the expressive power of our model. Some corresponding works are available in the literature that study multi-parameter persistence (Xin et al., 2023; Mukherjee et al., 2024; Dey & Xin, 2021b; 2019b;a; 2021a; Botnan et al., 2024). We also leave this for future work.

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

## A  LIST OF NOTATIONS

- $G = (V, E)$: A graph with vertex set $V$ and edge set $E$
- $G_X$: Candidate rationale subgraph
- $G_\epsilon$: Candidate noise or less relevant part of the graph
- $G_X^*$: Oracle rationale subgraph
- $G_\epsilon^*$: Oracle noise or less relevant part of the graph
- $f_\phi : G \to [0,1]^{|E|}$: Filtration functional
- $\mathcal{F}(G)$: Graph filtration determined by $f$
- $\mathcal{F}(G_{\leq t})$: Subfiltration consisting of subgraphs with $f(e) \leq t$
- $\mathcal{F}(G_{\geq t})$: Subfiltration consisting of subgraphs with $f(e) \geq t$
- $\mathcal{T}$: Topological invariant (e.g., persistence diagram)
- $d_{\text{topo}}$: Topological discrepancy
- $d_{\text{B}}$: Bottleneck distance between persistence diagrams
- $d_{\text{GH}}$: Gromov-Hausdorff distance between graphs
- $d_{\text{wass}}$: 1-Wasserstein distance
- $h_\phi$: GNN model for prediction
- $\sigma$: Extraction function to separate graph $G$ into $G_X$ and $G_\epsilon$
- $\varphi$: Vectorization function for persistence diagrams
- $\mathbb{P}_{\text{prior}}$: Prior distribution on edge filtration
- $\mathcal{L}_{\text{prior}}$: Prior regularization term
- $\alpha, \beta, \gamma$: Hyperparameters for loss function components

## B  MISSING PROOFS

*Proof.* (Proof of Theorem 3.4) By the assumption we know that the first term can only be optimized by $G_X \geq G_X^*$. We just need to show that $d_{\text{topo}}$ is uniquely maximized by $G_X^*$ among those $G_X \geq G_X^*$. In other words, we could assume that we have already restricted $f_\phi$ to the region satisfying $f_\phi|_{E_X^*} > 0.5 + \delta$ (the partition threshold $t = 0.5$ is fixed).

For a given $G$ and a fixed partition $G_X \sqcup G_\epsilon$ determined by some $f_\phi$, let $p_0, p_1$ be the 0-th and 1-st persistence diagrams, and $q_0, q_1$ be the 0-th and 1-st persistence diagrams. Observe that the bottleneck distance between the 0-th persistence diagrams $d_{\text{B}}(p_0, q_0)$ is maximized when

$$f_\phi(e) = 1\{e \in G_X\}. \tag{8}$$

The reason is that since we only care about edge filtrations, the filtration values on nodes can be viewed as some global minimum constant value which is commonly set to be time 0 (or more precisely, 1 for $G_X$ and 0.5 for $G_\epsilon$ since we build the filtration in the reversed ordering of importance). Then since $|E_\epsilon| > |E_X| \implies |q_0| > |p_0|$, we hope to maximize the death times of points in $q_0$ and minimize the death times of points in $p_0$ to maximize $d_{\text{B}}(p_0, q_0)$, which gives us the constant filtration function $f_\phi(e) = 1\{e \in G_X\}$ on each partition. Then, for constant filtration functions, the induced graph filtrations are essentially reduced to static graphs, and in consequences, persistent homology is essentially reduced to homology. For 0-degree homology, we just need to compare the 0-th Betti numbers $\beta_0^\epsilon$ and $\beta_0^X$ between $G_\epsilon$ and $G_X$. In that case, $d_{\text{B}}(p, q) = C(\beta_0^\epsilon - \beta_0^X) = C(|E_\epsilon| - |E_X|) = C(|G_E| - 2|E_X|)$ for some constant $C$ independent of $\phi$ or $G$. This is maximized when $G_X = G_X^*$.

The rest is to check the bottleneck distance $d_{\text{B}}(p_1, q_1)$ on 1-th persistence diagrams. In a similar way one can check that $d_{\text{B}}(p_1, q_1)$ should be maximized for some constant filtration function. Then the problem is again reduced to compare the 1-degree homology between $G_X$ and $G_\epsilon$. That is $|\beta_1^X - \beta_1^\epsilon|$. However, observe that $|\beta_1^X - \beta_1^\epsilon| \leq \beta_1$ for $\beta_1$ be the 1-st Betti number of the original graph. By the property of the Euler characteristic on a connected graph we know that $\beta_1 \leq |E| - |V| + 1 \leq |E| \leq |V|^2$. Therefore, $d_{\text{B}}(p_1, q_1) \leq M$ for some large enough $M$ over the whole dataset.

Based on that, since $d_{\text{topo}}$ is essentially a weighted sum of $d_{\text{B}}$ on both 0-th and 1-st persistence diagrams, we just need a large enough constant scaling factor on 0-th persistence diagrams. Then it can been guaranteed that our $d_{\text{topo}}$ is optimized by $G_X^*$ with $f_\phi^*(e) = 1\{e \in G_X^*\}$. Such constant factor can be easily learned by our neural networks, or fixed by hand in the model. $\qquad\square$

**Learnable Vectorization of Persistence Diagrams**: We need a collection of Lipschitz continuous functions on the space of persistence diagrams to some Euclidean space on which we can easily compute the expectation of marginal distributions. Such techniques are well studied as vectorization methods in topological data analysis. Here we apply a learnable vectorization function introduced by Hofer et al. (2019) to represent persistence diagrams as some $k$-dimensional vectors. The core idea is to learn $k$ parameterized kernels (e.g., exponential) to represent the distributions of points on the persistence diagrams. Each kernel, in that paper called structure element, is proved to be Lipschitz continuous with some constant $C$. Here we use a so-called Rational hat structure element given by

$$\varphi(p; \boldsymbol{c}, \boldsymbol{r}) = \sum_{\boldsymbol{x} \in p} \frac{1}{1 + \|\boldsymbol{x} - \boldsymbol{c}\|_2} - \frac{1}{1 + |\, |\boldsymbol{r}| - \|\boldsymbol{x} - \boldsymbol{r}\|_2\,|} \tag{9}$$

where $\boldsymbol{c}$ and $\boldsymbol{r}$ are learnable center and radii. Then $\Psi = \frac{1}{C}\varphi$, gives us a 1-Lipschitz continuous function.

## C    More Details about the Experiments

### C.1    Datasets

**Mutag** (Kazius et al., 2005): The dataset involves a task of predicting molecular properties, specifically determining whether a molecule is mutagenic. The functional groups -NO2 and -NH2 are regarded as definitive indicators that contribute to mutagenicity, as noted by (Luo et al., 2020b).

**Benzene** (Sanchez-Lengeling et al., 2020): The dataset comprises 12,000 molecular graphs sourced from ZINC15 (Sterling & Irwin, 2015). The objective is to identify the presence of benzene rings within a molecule. The carbon atoms in these benzene rings serve as the ground-truth explanations.

**BA-2Motifs** (Luo et al., 2020b): The dataset involves a binary classification task in which each graph combines a Barabasi-Albert base structure with either a house motif or a five-cycle motif. The graph's label and ground-truth explanation are based on the motif it includes.

**SPmotif** (Wu et al., 2022): The dataset consists of graphs that merge a base structure (such as a Tree, Ladder, or Wheel) with a motif (either a Cycle, House, or Crane). Each graph is manually infused with a spurious correlation between the base and the motif. The graph's label and the ground truth explanation are determined by the motif it contains.

**BA-HouseGrid**: The house and grid motifs are chosen because they do not have overlapping structures, such as those found in the house and $3 \times 3$ grid.

**BA-HouseAndGrid** (Bui et al., 2024): Each graph is based on a Barabasi-Albert structure and may be linked with either a house motif or a grid motif. Graphs that contain both types of motifs are labeled as 1, while those containing only one type are labeled as 0. Note that each motif appears at most once in each graph.

**BA-HouseOrGrid** (Bui et al., 2024): Similar to BA-HouseAndGrid, graphs that contain either house motif or grid motif are labeled as 1, while those containing neither type are labeled as 0. Note that each motif appears at most once in each graph.

**BA-HouseOrGrid-nRnd**: Similar to BA-HouseOrGrid, graphs that contain either n house motifs or n grid motifs are labeled as 1, where n is a random integer between 1 (inclusive) and n (inclusive). More formally:

- **Label Assignment**:

$$P(\text{Label} = 1) = 0.5, \quad P(\text{Label} = 0) = 0.5$$

- **For Label = 1**: Given $n \in \mathbb{Z}^+$, for each $i \in \{1, 2, \ldots, n\}$, the three possible manifestations are:

$$P(i \times \texttt{grid} + i \times \texttt{house}) = \frac{1}{6n},$$

$$P(i \times \texttt{grid}) = \frac{1}{6n},$$

$$P(i \times \texttt{house}) = \frac{1}{6n}.$$

When `grid` and `house` appear simultaneously, their counts are equal.

## C.2    Details on Hyperparamter Tuning

### C.2.1    Backbone Models

**Backbone Architecture.** We use a two-layer GIN (Xu et al., 2019) with 64 hidden dimensions and 0.3 dropout ratio for all baselines. We use a three-layer CINpp (Giusti et al., 2023) with 64 hidden dimensions and 0.15/0.3 dropout ratio for TOPING. For all datasets, we directly follow (Giusti et al., 2023) using enhanced Topological Message Passing scheme including messages that flow within the lower neighbourhood, the upper neighbourhood and boundary neighbourhood of the underlying cell complex. Considering that the largest chordless cycle for most interpretable motifs is equal to 5 (the BA-2Motifs dataset includes a 5-cycle, while most of the other motifs have chordless cycles with a maximum length of 4), we lift the maximum length of a chordless cycle to 5 as the cell(dim=2).

**Data Splits.** For BA synthetic datasets, we follow the previous work (Miao et al., 2022; Chen et al., 2024; Bui et al., 2024) to split them into three sets(80%/10%/10%). For SPmotifs and real datasets, we use the default splits.

**Evaluation.** We report the performance of the epoch with the highest validation prediction accuracy and use these models as the pre-trained models. If multiple epochs achieve the same top performance, we choose the one with the lowest validation prediction loss.

## C.3    Interpretation Visualization

We provide visualization of the learned interpretabel subgraphs by GSAT and TOPING in the different datasets. The transparency of the edges shown in the figures represents the normalized attention weights learned by interpretable method. Note that we no longer need min-max normalization like (Miao et al., 2022) for better visualization, we can directly use edge attention to visualize through rational filtration learning, because persistent homology gap has guaranteed that our edge attention is easy to be distinguished.

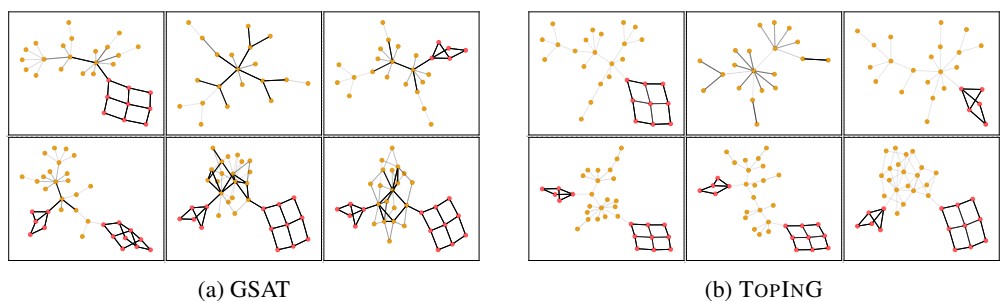

(a) GSAT                                        (b) TOPING

Figure 5: Learned interpretable subgraphs by GSAT and TOPING on BA-HouseAndGrid. Nodes colored pink are ground-truth explanations.

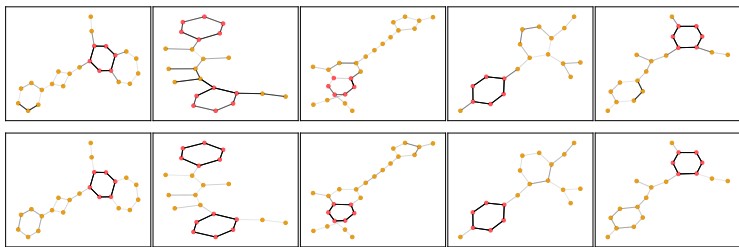

Figure 6: Visualizing attention of GSAT (first row) and TOPING (second row) on Benzene. Nodes colored pink are ground-truth explanations.

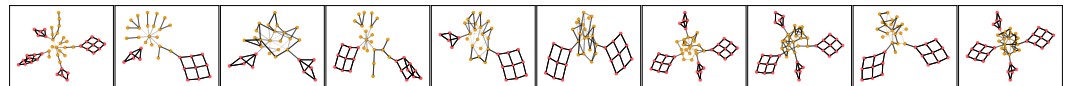

Figure 7: The rationals of BA-HouseOrGrid-2Rnd learned by TOPING. Nodes colored pink are ground-truth explanations.

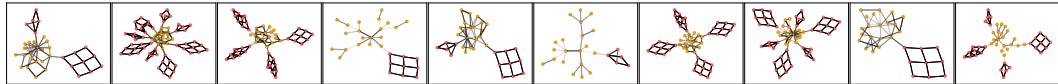

Figure 8: The rationals of BA-HouseOrGrid-4Rnd learned by TOPING. Nodes colored pink are ground-truth explanations.

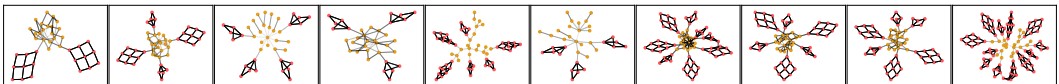

Figure 9: The rationals of BA-HouseOrGrid-6Rnd learned by TOPING. Nodes colored pink are ground-truth explanations.

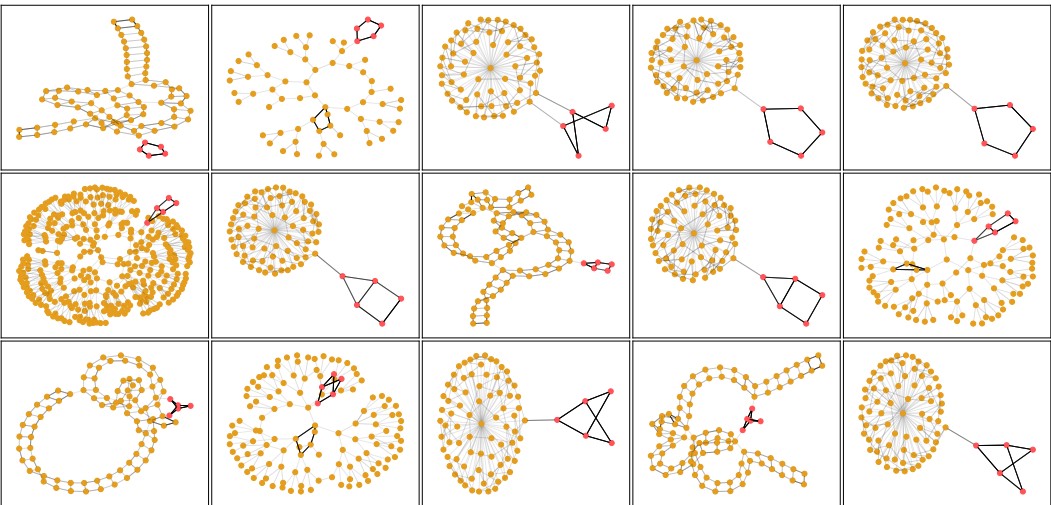

Figure 10: The rationals of SPmotif0.9 learned by TOPING. Nodes colored pink are ground-truth explanations.

