# OpenReview forum: "TopInG: Topologically Interpretable Graph Learning via Persistent Rationale Filtration"
_ICLR.cc/2025/Conference — Submitted to ICLR 2025_

### Official Review · Reviewer_xsRV · 2024-10-23

**Soundness:** 2
**Presentation:** 2
**Contribution:** 2
**Rating:** 6
**Confidence:** 4

**Summary:**

This paper proposes a framework TOPING for XGNN, which is similar to the GSAT pipeline. It introduces a rationale filtration learning technique and a topological discrepancy constraint to separate important subgraphs from random ones. The method offers theoretical guarantees and improves both accuracy and interpretability, outperforming existing approaches by over 20% in experiments. However, the comparison with other methods is potentially unfair due to differences in evaluation metrics or datasets.

**Strengths:**

1. This paper introduces an existing method based on the assumption, "either explicitly or implicitly that the subgraph rationales are nearly invariant across different instances within the same category of graphs." This is a reasonable and novel aspect of the problem. However, there are few experiments to support this idea.
2. This paper introduces the $p$-homology functor to XGNN, which is a novel approach to addressing this problem.

**Weaknesses:**

1. Some expressions are confusing. For example, in Equation 3, the meaning of $\mathcal{R}$ is unclear. In Figure 2, the meaning of $L_{topo}$ is also not explained.
2. In subsection 3.2, the distribution of the prior is determined by the hyperparameters $\mu$ and $r$. However, $\mu$ is a pre-defined parameter, raising the question of how to set this parameter. Additionally, $r$ is a learnable parameter. Will this setting lead to a trivial solution, where the divergence between $\mathcal{N}(\mu_1,r_1)$ and $\mathcal{N}(\mu_2,r_2)$ is small?
3. The comparison is potentially unfair. In Appendix C.2.1, all baselines are based on the GIN backbone, while TOPING uses the CINpp backbone, which could result in biased comparisons.

**Questions:**

1. In Figure 1, the meaning of the row lines is unknown.

---

> ### Author Response · Authors · 2024-11-25
> **Initial Response 1**
>
> We appreciate the reviewer's feedback.
>
> **Weakness 1 and Question 1:**
>
> [Answer:]
> We have clarified our notation and visualizations:
> - $\mathcal{R}$ in Equation 3 represents the cross-entropy loss between predicted label $\bar{y}_G$ and ground truth label $y_G$
> - The topological loss $L_{topo}=d_{topo}$ is implemented through our topological discrepancy measure
> - In Figure 1, the horizontal lines form a persistent barcode -- a complete discrete representation of persistent homology. Each line represents a basic cycle's lifecycle (birth---death) in the graph filtration
>
> For more detailed explanations of notation and TDA concepts, please refer to [pdf link](https://anonymous.4open.science/r/TopoEx-1EE2/OpenReview/rebuttal_update.pdf).
>
> **Weakness 2**
> - Regarding the setting of the hyperparameter $\mu$, please refer to our answer to the second common concern in the general response.
> - Regarding the learnable parameter $r$: We minimize the KL-divergence between the learned edge distribution and the prior distribution, or equivalent maximize the log-likelihood.
> Since $\mu_1$ and $\mu_2$ are fixed, the log-likelihood is enlarged when the edge distribution are more concentrated around those two means $\mu_1$ and $\mu_2$, which also increases the divergence between $\mathcal{N}(\mu_1, r_1)$ and $\mathcal{N}(\mu_2, r_2)$. The only degenerate case we need to prevent is the mode collapse where all points cluster around a single center.
> We add a simple penalty term to penalize very small variance.
> While more sophisticated approaches exist, this straightforward solution works effectively for our model.

---

> > ### Author Response · Authors · 2024-11-25
> > **Initial Response 2**
> >
> > **Weakness 3:** Regarding comparison
> >
> > [Answer:] The baseline methods are not compared with CIN backbone network due to their significantly degraded performance with CIN. We observe that the training becomes highly unstable -- interpretation scores (AUC) fail to converge despite prediction score (ACC) convergence, and all edge attentions cluster around constants. See [Figure 1](https://anonymous.4open.science/r/TopoEx-1EE2/OpenReview/Figure%201.png) for illustration. Even we record the best performance achieved across all epochs as an upper bound approximation (marked as CINpp* in Table 1), the results remain notably inferior to their GIN counterparts, showing both worse means and larger variances.
> >
> >
> > Table 1:
> >
> > | method | Backbone | BA-2motifs | BA-HouseOrGrid-2Rnd | Mutag |
> > |--------|----------|------------|-------------------|--------|
> > | GSAT | GIN | 98.74 ± 0.55 | 76.02 ± 3.64 | 99.38 ± 0.25 |
> > | | CINpp* | 91.12 ± 4.93 | 75.98 ± 7.09 | 97.27 ± 0.47 |
> > | GMT-Lin | GIN | 97.72 ± 0.59 | 70.45 ± 5.75 | 99.87 ± 0.09 |
> > | | CINpp* | 91.03 ± 5.24 | 67.91 ± 5.10 | 97.48 ± 0.81 |
> > | TopInG | GIN | 99.57 ± 0.60 | 88.74 ± 1.70 | 95.79 ± 1.93 |
> > | | CINpp | 100.00 ± 0.00 | 100.00 ± 0.00 | 96.38 ± 2.56 |
> >
> >
> >
> >
> > While we are still investigating why baseline methods struggle with CIN backbone networks, we could share some initial observations based on the concrete loss function (Equation (8) and (9) in [1]) of GSAT, shown as follows:
> >
> >  &nbsp;&nbsp;&nbsp;&nbsp;&nbsp;&nbsp;&nbsp;&nbsp;&nbsp; &nbsp;&nbsp; $Loss=\min_{\phi,\theta} -\mathbb{E}[\log \mathbb{P}_\theta(Y|G_S)]$
> >
> >  &nbsp;&nbsp;&nbsp;&nbsp;&nbsp;&nbsp;&nbsp;&nbsp;&nbsp; &nbsp;&nbsp; &nbsp;&nbsp;&nbsp;&nbsp;&nbsp;&nbsp;&nbsp;&nbsp;&nbsp; &nbsp;&nbsp; $ + \beta\mathbb{E}[\text{KL}(\mathbb{P}_\phi(G_S|G)||\mathbb{Q}(G_S))]$
> >
> > where $\text{KL}(\mathbb{P}_\phi(G_S|G)||\mathbb{Q}(G_S))$ is realized as follows
> >
> >  &nbsp;&nbsp;&nbsp;&nbsp;&nbsp;&nbsp;&nbsp;&nbsp;&nbsp; &nbsp;&nbsp; $\text{KL}(\mathbb{P}_\phi(G_S|G)||\mathbb{Q}(G_S)) $
> >
> > &nbsp;&nbsp;&nbsp;&nbsp;&nbsp;&nbsp;&nbsp;&nbsp;&nbsp; &nbsp;&nbsp;&nbsp;&nbsp;&nbsp;&nbsp;&nbsp;&nbsp;&nbsp;&nbsp;&nbsp; &nbsp;&nbsp;  $= \sum_{(u,v)\in E} p_{uv} \log \frac{p_{uv}}{r}$
> >  $ + (1-p_{uv}) \log \frac{1-p_{uv}}{1-r} $
> >
> >
> > with some prefixed $r\in [0,1]$ chosen to be $0.7$ in [1].
> >
> >
> >  One observation is that:
> >  - when the prediction network becomes sufficiently powerful in the sense that it can easily fit the data fast, then there exists an optimal solution with all edge attention values are trivially identical, causing interpretation to fail.
> >
> >  Similar observations appear in GMT[2]. This phenomenon is reminiscent of a common issue of GAN training[4], where an overly powerful discriminator can prevent effective generator training.
> >
> >  Based on this, we hypothesize that:
> >  - baseline methods relying on a single threshold value to control rationale subgraph complexity are more susceptible to learning trivial interpretation solutions when paired with more powerful or complex prediction models.
> >
> > This also aligns with previous findings that GAT sometimes significantly underperforms GIN and GCN on interpretation tasks[3].
> >
> >
> >
> >
> > Reference:
> >
> > [1] Miao, Siqi, et al. "Interpretable and generalizable graph learning via stochastic attention mechanism." International Conference on Machine Learning. PMLR, 2022.
> >
> > [2] Chen, Yongqiang, et al. "How Interpretable Are Interpretable Graph Neural Networks?." arXiv preprint arXiv:2406.07955 (2024).
> >
> > [3] Ngoc Bui, Hieu Trung Nguyen, Viet Anh Nguyen, Rex Ying, "Explaining Graph Neural Networks via Structure-aware Interaction Index", Proceedings of the 41st International Conference on Machine Learning, PMLR 235:4854-4883, 2024.
> >
> > [4] Goodfellow, I., et al. "Generative Adversarial Networks." NeurIPS (2014)

---

> ### Comment · Reviewer_xsRV · 2024-11-25
> **Response to Weakness 2**
>
> Thanks for the explanation and experiments.
>
> For the Weakness 2, would you like to present the final $r_1$ and $r_2$?  In the global response,  "The key insight is that our prior regularization acts as a clustering mechanism of edges into two clusters on [0,1]", this sentence is suitable for all explainers. For my understanding, GSAT also uses a Marginal Distribution as the prior. It would be helpful to find out the difference between yours and GSAT.

---

> ### Author Response · Authors · 2024-11-25
> **Follow-up Response**
>
> Thank you for the follow-up comments.
>
> **Regarding $r_1$ and $r_2$**
> - [Answer:] We checked our model train on the dataset BA-HouseOrGrid-2Rnd as an example, the learned $(r_1, r_2)=(0.2144,0.2167)$.
> We also uploaded two plots of some example distributions of edge filtrations.
>     - [Fig1](https://anonymous.4open.science/r/TopoEx-1EE2/OpenReview/toping_edge_singleplot.png) is one example distribution of the edge filtration values.
>     - [Fig2](https://anonymous.4open.science/r/TopoEx-1EE2/OpenReview/toping_edge_avgplot.png) is an average 10 distributions of edge filtration values on 10 sample graphs.
>
> **Regarding the question about prior regularization and comparison with GSAT:**
> - [Answer:] We appreciate the insightful observation about clustering mechanisms in explainable GNN models. While there are surface similarities between our prior regularization and GSAT's approach, there are fundamental differences in both design and functionality:
>
>     - Role of Prior Regularization:
>         In GSAT, the information bottleneck loss (info-loss) serves as the primary mechanism for identifying rationale subgraphs. We consider that the "workload" of this info-loss extends significantly beyond a simple clustering mechanism. As such, its parameter $r$ requires careful consideration during training (including techniques like warm-up or gradual decay). In contrast, our topological discrepancy loss is the primary driver for interpretation, with the prior regularization playing a supportive clustering role.
>     - Architectural Differences:
>         Our bimodal Gaussian prior naturally encourages separation between rationale and non-rationale edges without requiring precise parameter tuning. The two modes at $μ_1 = 0.25$ and $μ_2 = 0.75$ provide clear separation points, while the learnable variances $(r1,  r2)$ allow flexibility in the clustering boundary.
>     - Stability and Robustness:
>         Because our model's interpretability primarily stems from the topological discrepancy loss, it is less sensitive to the exact values of the prior regularization parameters, which makes the training process stable and robust across different datasets, without careful hyperparameter tuning.
>
> - While investigating the broader implications of bimodal or even multi-modal versus unimodal priors in interpretable GNNs would be fascinating, it extends beyond the scope of this work. We consider one of our main contributions is introducing topological methods to this domain, and we believe this will benefit interpretable GNN problems in general.

---

> > ### Comment · Reviewer_xsRV · 2024-11-26
> > **Thanks for the Response**
> >
> > Thanks for the response. I don't have any other question.

---

> > > ### Author Response · Authors · 2024-11-26
> > > **Thanks for the review**
> > >
> > > Thank you for the time and careful review. We appreciate all the constructive feedback and insightful comments to help us improve both the clarity and quality of our manuscript.

---

### Official Review · Reviewer_rSzd · 2024-10-24

**Soundness:** 2
**Presentation:** 2
**Contribution:** 3
**Rating:** 6
**Confidence:** 2

**Summary:**

This paper proposes an interpretable GNN framework that leverages techniques from topological data analysis, specifically persistent homology. The method splits an input graph into a rationale subgraph (important subgraph) and a noise subgraph, where the rationale subgraph is used for both predicting the target and providing an explanation for the prediction. The authors introduce a novel loss function that quantifies the topological discrepancy between the rationale and noise subgraphs and jointly optimize this loss along with the prediction loss. The framework is evaluated using several synthetic datasets and two real-world molecular datasets, demonstrating its effectiveness.

**Strengths:**

1.	This paper introduces an innovative use of topological analysis to provide intrinsically explainable GNNs, offering a promising direction for future research.
2.	The authors proved that the method can optimize to the true important subgraph under certain conditions, providing a solid mathematical foundation for the approach.

**Weaknesses:**

1.	The writing is difficult to follow due to unclear definitions of the mathematical symbols used (see questions below). The figures are not well-explained and can be confusing, especially for readers with limited knowledge of topological analysis.
2.	The experimental datasets are limited, focusing only on synthetic and relatively simple molecular datasets. Additionally, qualitative visualizations for the MUTAG dataset are missing, which reduces the clarity of the results.

**Questions:**

1. What is $\beta_1$ in Fig. 1?
1. In Eq. 3, what is $\mathcal R$?
2. What is the difference between $f_{\phi}, h_{\phi}$? In Fig.2 they are all equal to $GNN_\phi$ so they are the same? And is $\phi$ the parameters of GNN?
3. After reading Sec 3 I still have problem understanding how the model predicts. In Fig.2 it seems that the model first generates a subgraph $G_X$ according to the edge score from $f_\phi(G)$, feed it into a GNN $h_\phi$, somehow concatenate with $\mathcal T_X, \mathcal T_\epsilon$, and use a classifier to get the final prediction. What is $\mathcal T_X, \mathcal T_\epsilon$ and how are they concatenated with output of GNN $h_\phi$? A more detailed explanation of Fig.2 is needed.
4. Row 267-269:the defined $\mathcal G, \mathcal F(\mathcal G)$ are never used.
5. Does the measure of topological discrepancy also consider the node/edge features? It would be interesting to see how the method performs on graphs where not only topological structure but also node/edge features are relevant.

---

> ### Author Response · Authors · 2024-11-25
> **Initial Response**
>
> We appreciate the reviewer's constructive feedback.
>
> **Weakness 1, Question 2,3,4:**
> - Regarding the preliminary introduction to TDA, mathematic definitions,notations in equation (3) and clarification of loss functions and the architecture of our model, please refer to our answer to the first common question in the general response for a complete revision. Here we give some short answers for spedific clarification:
> - [Answer to Q2:]  $\mathcal{R}$ in the origical Eq. 3 represents the standard cross-entropy loss between the predicted label $\bar{y}_G$ and the ground truth label $y_G$. We have revised the equation as Eq. 2 in Section 3 in [pdf link](https://anonymous.4open.science/r/TopoEx-1EE2/OpenReview/rebuttal_update.pdf).
> - [Answer to Q3:] Strictly speaking, while $f_\phi$ and $h_\phi$ aren't identical, they share a substantial common component in the backbone $GNN_\phi$. Specifically:
>
>     - $GNN_\phi(G)$ processes the input graph to produce permutation-equivalent representations (node/edge embeddings)
>     - $f_\phi = MLP_f \circ h_\phi$ reduces $GNN_\phi(G)$'s output to a 1-dimensional permutational equivalent representation via a multi-layer perceptron $MLP_f$
>     - $h_\phi = MLP_h \circ Pool \circ GNN_\phi$ applies pooling to $GNN_\phi(G)$ for graph-level representation, followed by prediction MLP
>
>     Here we omit the details of parameters in $MLP$ models for simplicity. For a complete discussion, please see Section 3 in [pdf link](https://anonymous.4open.science/r/TopoEx-1EE2/OpenReview/rebuttal_update.pdf).
>
> - [Answer to Q4:]
> $T_X$ and $T_\epsilon$
> are persistent homology representations on filtrations of $G_X$ and $G_\epsilon$ respectively.
> These permutation-invariant graph representations are concatenated with the pooled graph representation output by $Pool \circ GNN_\phi$. Then they are altogether fed into the final classifier $MLP_h$.
> Please
> refer to the last paragraph in Section 3 in [pdf link](https://anonymous.4open.science/r/TopoEx-1EE2/OpenReview/rebuttal_update.pdf) for a more detailed explanation of the loss function and architecture of our model.
>
> **Weakness 2:** Regarding datasets.
> - We have added some example results of MUTAG dataset in [pdf link](https://anonymous.4open.science/r/TopoEx-1EE2/OpenReview/mutag_TopInG.pdf). We have also added more illustrative example results on different datasets in [pdf link](https://anonymous.4open.science/r/TopoEx-1EE2/OpenReview/MoreInterpretationVisualization.pdf).
> - We acknowledge the limited availability of benchmark datasets for interpretable GNNs in general, as they require both graph and rationale subgraph labels. Our evaluation uses most of the standard benchmark datasets employed by baseline methods, most of which are synthetic or semi-synthetic. Creating comprehensive benchmark datasets would be valuable future work for advancing this field.
>
>
>
> **Question 1** Regarding $\beta_1$ in Fig. 1:
> - [Answer:] $\beta_1$ here indicates the dimension of each 1-st homology vector space $H_1(G_{\leq t})$, presented in the boxes under each subgraph. Essentially, $\beta_1$ counts the number of basic cycles in the graph. In the revised version of the Fig. 1 in [pdf link](https://anonymous.4open.science/r/TopoEx-1EE2/OpenReview/rebuttal_update.pdf), we have replaced it with $H_1$ for clarity. We have explained them in the revised caption.
>
> **Question 5:** Regarding unused notations.
> - [Answer:] Yes, we have removed them in the version.
>
> **Question 6** Regarding node/edge features
> - [Answer:] Yes, TopInG is able to takes node/edge features into account. In the real datasets Mutag and Benzene, each atom (node) and bond (edge) of the chemical molecules has distinct features derived from AtomEncoder and BondEncoder (provided by OGB). TopInG utilizes these actual features to obtain importance scores for each atom and molecule through rationale filtration learning. For the Mutag dataset, the functional groups -NO2 and -NH2 can be differentiated using TopInG as indicators, in contrast to -CH2 or -CO2, which share a similar topological structure with the ground-truth mutagens. In the case of Benzene, the carbon atoms in the benzene rings can also be identified using TopInG, distinguishing them from a random ring of six atoms.

---

> > ### Author Response · Authors · 2024-11-25
> >
> > We appreciate your thorough review and hope we have adequately addressed all your concerns. If you have any additional questions or would like further clarification on any point, we welcome continued discussion.

---

> > ### Comment · Reviewer_rSzd · 2024-11-25
> >
> > Thanks for the additional explanation and improvement of the notations, which are helpful for me to understand your method.
> >
> > However, Fig. 2 requires a careful and detailed revision. For instance, the term ‘clf’ is never mentioned in the paper and might be better replaced with $MLP_h$ for consistency. Additionally, the equation $h_\phi = MLP_h \cdots GNN_\phi$ implies that it is incorrect to denote $h_\phi = GNN_\phi$ in the figure.
> >
> > Furthermore, the pooling function $Pool$ would benefit from a more explicit definition to enhance clarity.

---

> > > ### Author Response · Authors · 2024-11-26
> > > **Follow-up Response**
> > >
> > > **Answer**:
> > >
> > > Thank you for the follow-up comments and suggestions regarding Figure 2. We appreciate the reviewer's careful review that has helped us identify several areas needing clarification.
> > >
> > > We have made the following updates to Figure 2 to improve clarity and maintain consistency:
> > >
> > > - Replaced 'clf' with $MLP_h$ for better alignment with the mathematical notation used in the paper.
> > > - Removed the notation $h_\phi$ and instead explicitly presented its components: $GNN_\phi$, $Pool$, and $MLP_h$.
> > > - Added clarification about the pooling layer: we use a standard pooling function (e.g., sum-pooling) to transform the permutation equivariant representation to a permutation invariant graph representation.
> > > - Revised both the model architecture description in the main text and the figure caption to ensure consistent notation and terminology throughout.
> > >
> > > The updates can be checked in [Fig2](https://anonymous.4open.science/r/TopoEx-1EE2/OpenReview/model_architecture_revise.png) and [PDF](https://anonymous.4open.science/r/TopoEx-1EE2/OpenReview/rebuttal_update.pdf). We believe these revisions improve notation consistency and functional clarity.
> > >
> > > Please let us know if there are any additional questions or concerns. We welcome continued discussion to ensure the presentation of our work is as clear and precise as possible.

---

> > > > ### Comment · Reviewer_rSzd · 2024-11-26
> > > >
> > > > Thanks for the update. I don’t have further question.

---

> > > > > ### Author Response · Authors · 2024-11-26
> > > > > **Thanks for the review**
> > > > >
> > > > > Thanks again for your time and careful review. If you think we have solved all your concerns, we would kindly ask you to consider reassessing the rating.

---

### Official Review · Reviewer_gpKc · 2024-10-31

**Soundness:** 2
**Presentation:** 1
**Contribution:** 2
**Rating:** 5
**Confidence:** 4

**Summary:**

The paper introduces TOPING, a novel intrinsically interpretable framework for GNNs that uses topological data analysis (TDA) to identify stable rationale subgraphs. TOPING addresses key limitations in existing interpretable GNN models by using a rationale filtration learning approach. Extensive experimentation shows improvements over state-of-the-art interpretability and predictive performance.

**Strengths:**

S1. Using TDA for intrinsically interpretable GNNs is interesting.

S2. The authors provide theoretical guarantees to show that topological discrepancy optimally captures rationale subgraphs.

S3. The authors present extensive experiments across multiple datasets.

**Weaknesses:**

W1. The mathematical notation and presentation could be improved, as some definitions and descriptions are unclear or ambiguous. Clarifying these elements would enhance the paper’s readability for readers less familiar with topological data analysis.

W2. The paper would benefit from a detailed time complexity analysis and runtime study, particularly in comparison with baseline methods. This addition would clarify whether TOPING’s performance improvements come at the cost of efficiency.

W3. Since TOPING requires learning a filtration function that assigns values to each edge, this method may become impractical for large graphs due to the need to compute and store $|E|$ values. Further discussion on how TOPING might be adapted or approximated for larger graphs would be valuable.

**Questions:**

Q1. The explanation of TDA in Section 2.2 is unclear. In Figure 1, the subgraph $G_{t}$ appears to have more edges as $t$ decreases. However, in line 170, it seems that $G_{t}$ should contain more edges as $t$ increases. Could the authors clarify this apparent discrepancy?

Q2. In line 271, the term "topological invariants $\tau$ " is introduced without a clear mathematical definition. Could the authors provide a formal definition for $\tau$ to improve understanding?

Q3. In line 278, Equation (3) is not properly referenced. Besides, it appears disconnected from the context. Could the authors clarify its relevance here?

Q4.  The authors state that $f_\phi$ and $h_\phi$ share parameters. However, $f_\phi$ is defined as a function mapping to $[0,1]^{|E|}$, while $h_\phi$ seems to map to $\mathbb{R}^{|V|}$. Could the authors explain how parameter sharing is feasible between these two functions with distinct output spaces?

---

> ### Author Response · Authors · 2024-11-25
> **Initial Response 1**
>
> We appreciate the reviewer's constructive feedback.
>
> **Weakness1**
> Regarding the preliminary introduction to TDA: Please refer the first answer to the first common concern in the general response.
>
> **Weakness2**
> Regarding time complexity analysis and runtime study.
>
> - [Answer:]
> The theoretical time complexity of the computation of persistent homology on graphs is known to be relatively efficient - $O(n\log n)$ where $n$ is the total number of nodes and edges in the graph [1,2]. However, we acknowledge that our current implementation is slower than the fastest baseline methods (5X to 20X slower). This performance gap stems primarily from the lack of optimized GPU implementations for persistent homology computation, which we view as an important open challenge for both the TDA and ML communities.
> - Importantly, we also want to note that since ground truth rationale subgraphs are never used during training or validation, the interpretation performance cannot be artificially improved simply through larger parameter spaces or longer inference times.
>
>
> **Weakness 3:**
> Regarding edge filtration learning and large graphs in practice.
>
> - [Answer:]
> First, it is worth noting that all interpretable baseline methods also learn and store 1-dimensional edge embeddings as filtration functions.
> - For larger graphs, we acknowledge the computational challenges of TDA-based methods, as mentioned above. We see two promising directions for improvement:
> (1) Development of efficient GPU implementations.
> (2) Using carefully designed GNN models to approximate persistent homology computations [3]
>
>
>
> **Q1:** Figure 1 Clarification.
> - [Answer:] We have revised both the TDA preliminary introduction and Figure 1 (see our response to common concerns).
> Regarding the discrepancy between descending t values in Figure 1 and ascending t values in line 170: In our model, $t=1-f$ since we add edges from higher to lower importance, where importance is measured by filtration value $f(e)$. Thus, while filtration values $f(e)$ decrease in our graph filtration, t increases. We've updated Figure 1 to use $f(e)$ instead of t to avoid confusion.
>
>
> **Q2 and Q3:** Regarding the definition of topological invariant and our loss function in Equation (3).
> - [Answer:] The topological invariant refers to our persistent homology representations. We have provided clearer definitions of both topological invariant and topological discrepancy (equation (1)), with accompanying explanations. Please see Section 3 in [pdf link](https://anonymous.4open.science/r/TopoEx-1EE2/OpenReview/rebuttal_update.pdf)) for complete details.
>
>
>
> **Q4:** Regarding the confusion of the shared parameters between $f_\phi$ and $h_\phi$.
>
> - [Answer:] While $f_\phi$ and $h_\phi$ aren't identical, they share a substantial common component in the backbone $GNN_\phi$. Specifically:
>
>     - $GNN_\phi(G)$ processes the input graph to produce permutation-equivalent representations (node/edge embeddings)
>     - $f_\phi = MLP_f \circ h_\phi$ reduces $GNN_\phi(G)$'s output to a 1-dimensional permutational equivalent representation via a multi-layer perceptron $MLP_f$
>     - $h_\phi = MLP_h \circ Pool \circ GNN_\phi$ applies pooling to $GNN_\phi(G)$ for graph-level representation, followed by prediction MLP
>
>     Here we omit the details of parameters in $MLP$ models for simplicity. For a complete discussion, please see Section 3 in [pdf link](https://anonymous.4open.science/r/TopoEx-1EE2/OpenReview/rebuttal_update.pdf)).
>
> Reference
>
> [1] Edelsbrunner, H., Letscher, D., & Zomorodian, A. (2000). Topological Persistence and Simplification. Discrete & Computational Geometry, 28, 511-533.
>
> [2] Tamal Krishna Dey and Yusu Wang. Computational Topology for Data Analysis. Cambridge University Press, 2022.
>
> [3] Zuoyu Yan, Tengfei Ma, Liangcai Gao, Zhi Tang, Yusu Wang, and Chao Chen. Neural approximation of graph
> topological features. In Proceedings of the 36th International Conference on Neural Information Processing Systems,
> NIPS ’22, Red Hook, NY, USA, 2022. Curran Associates Inc. ISBN 9781713871088.

---

> ### Author Response · Authors · 2024-11-25
>
> We appreciate your thorough review and hope we have adequately addressed all your concerns. If you have any additional questions or would like further clarification on any point, we welcome continued discussion.

---

> ### Author Response · Authors · 2024-12-02
>
> Dear Reviewer gpKc,
>
> As we approach the end of the discussion period, we would greatly appreciate your assessment of whether our detailed responses have fully addressed your concerns. If you feel your concerns have been satisfactorily resolved, we respectfully request that you consider updating your rating to reflect this. We remain available to address any remaining questions or provide additional clarification if needed.
>
> Thanks again for your review and consideration.
>
> Best regards,
>
> The Authors

---

### Official Review · Reviewer_nH1H · 2024-11-02

**Soundness:** 3
**Presentation:** 2
**Contribution:** 2
**Rating:** 6
**Confidence:** 3

**Summary:**

In this paper, the authors propose a graph explanation method based on the persistent homology to identify stable rational subgraphs via persistent rationale filtration learning. Specifically, they propose a topological discrepancy loss to achieve the goal. Experimental results seem to outperform state-of-the-art models in most datasets.

**Strengths:**

- The idea of using topological data analysis for graph explanation is novel and interesting.

- The derivation of the proposed method seems to be solid with theoretical insights.

**Weaknesses:**

- The preliminary, especially the TDA, is not very clear, making it hard for readers without the knowledge of persistent homology (I don’t think it is necessarily a prerequisite knowledge for graph explainability). What is the difference of $H$ between the $p$-homology functor $H_p (\mathcal{F} (G))$ and $H_p (G_{\le t})$? Figure 1 is also not easy to understand with a clear explanation of the homology groups. The caption is over lengthy and not illustrating the intuitive idea very well. It will be better to segment the long caption into several parts to make it more readable.

- In the introduction, they mention one challenge that exists in current graph explanation models: that different graphs may contain different core subgraphs even within the same category. Yet, I don’t see why the proposed method can tackle this problem well. The training procedure involves learning a filtration function to separate the graph. How such a mechanism can tackle the challenge and why the separating method in other methods cannot are not well discussed. It is recommended to add more analysis (or examples) to demonstrate why the proposed method can address this challenge and others cannot.

- The prior regularization seems to be empirical. The hyperparameters of the prior are not trivial. This raises concern whether the proposed method is sensitive to the selection of the prior and the corresponding parameters. More discussion is encouraged.

**Questions:**

- See the weakness above

- The results of MUTAG are not impressive in terms of both interprebility and accuracy. Further analysis for these results are required for better understanding the proposed method.

---

> ### Author Response · Authors · 2024-11-24
> **Initial response 1**
>
> We appreciate the reviewer's constructive feedbacks.
>
> **Weakness 1**
>
> Regarding the preliminary introduction to TDA, please refer to our answer to the first common concern in the general response.
>
> Regarding the different of $H$ between $H_p(\mathcal{F}(G))$ and $H_p(G_{\leq t})$:
> - $H$ is simply a notation representing homology, without any concrete meaning on its own. The key difference lies in what it's applied to:
>
>     - $H_p(G_{\leq t})$ is the homology group (vector space) of a single graph $G_{\leq t}$
>
>     - $H_p(\mathcal{F}(G))$ represents the persistent homology of a graph filtration $\mathcal{F}(G)$ (a chain of nested subgraphs of $G$). It is a richer algebraic structure consisting of:
>
>         1. A collection of homology vector spaces ${H_p(G_{\leq t})}$
>         2. Linear maps between these spaces ${H_p(G_{\leq t_1}) \to H_p(G_{\leq t_2})}$
>
>
> Regarding Figure 1, we have enhanced its clarity in several ways:
>
> - We've added more explanation of homology groups (vector spaces) in Appendix B.
> - We've restructured the lengthy caption into three more digestible parts:
>
>     1. The first part serves as an illustrative example of persistent barcode on a graph filtration, now included in the preliminary introduction to TDA (see the last part of the "Persistent Homology" paragraph in Section 2.1 in [pdf link](https://anonymous.4open.science/r/TopoEx-1EE2/OpenReview/rebuttal_update.pdf)).
>     2. The second part remains as a concise caption for Figure 1.
>     3. The third part appears at the beginning of the experiment section, demonstrating "what could be learned by our topological discrepancy" (see Section 4 in [pdf link](https://anonymous.4open.science/r/TopoEx-1EE2/OpenReview/rebuttal_update.pdf)).
>
> **Weakness 3**
>
> Regarding the hyperparameter settings of our prior regularization, please refer to our answer to the second common concern in the general response.

---

> ### Author Response · Authors · 2024-11-25
> **Initial Response 2**
>
> **Weakness 2**: regarding the challenge of varriform rational subgraphs, why our method works while other methods cannot.
>
> [Answer: ]
> Thank you for this insightful question. We will address how TopInG handles variiform rationale subgraphs from three perspectives: theoretical guarantees, intuitive understanding, and empirical validation.
>
> Theoretical Perspective
> - Previous methods implicitly or explicitly assume low variance in rationale subgraphs within each category. For instance, GSAT's theoretical guarantee (Theorem 4.1, (Miao et al., 2022)) assumes that the label $y_G$ is determined by an invertible function $y_G = f(G_X^*) + noise$, implying that $f^{-1}(y_G)$ must be nearly a delta-distribution with small noise. GMT (Chen et al., 2024) inherits this limitation from GSAT. DIR (Wu et al., 2022) is designed to be robust against background noise. It also assumes invariant causal structures (Theorem 2 and Corollary 1 in Appendix C). In contrast, TopInG's theoretical guarantees make no such restrictive assumptions, allowing it to handle variiform rationale subgraphs in theory.
>
> Intuitive Understanding
> - The key distinction lies in how rationale subgraphs are identified. Existing methods (GSAT, GMT, DIR) rely on global threshold parameters (e.g., r=0.7 for GSAT/GMT, r=0.4 for GMT) to control rationale subgraph size or information content. This approach works well when rationale subgraphs have low variance but struggles with high variability. In contrast, TopInG identifies rationale subgraphs through statistical and topological differences between the rationale and complement (noise) subgraphs across the entire dataset. This self-adjusting approach naturally accommodates variable rationale structures without requiring them to conform to specific size or complexity constraints. Also, as discussed in our previous answer to the concern regarding the hyperparameter choices of our prior regularization, the choice of $\mu$ in our prior regularization is very robust and enjoys great flexibility. Our model's optimality does not depend on a specific threshold value, which is very different from previous approaches.
>
> Empirical Validation
> - Our experiments on the BA-HouseOrGrid-nRnd dataset (Figure 3 in our original paper) demonstrate this difference empirically. As the number of possible rationale subgraphs increases, the performance of existing methods degrades significantly, while TopInG maintains robust performance. This validates our theoretical conjecture that methods depending on global thresholds struggle with variiform rationales, while our topological approach remains effective.
>
> **Question**: regarding the result of MUTAG
>
> [Answer:]
> - The results of MUTAG can be attributed to the uniquely simple structure of MUTAG's rationale subgraphs. MUTAG's rationale subgraphs consist of just two edges sharing a common node (the functional groups -NO2 and -NH2). This represents the simplest possible non-trivial subgraph structure, lacking the topological complexity present in other datasets. Specifically:
>
>     - There are no cycles (1-homology features)
>     - The 0-homology structure (connectivity) is nearly trivial
>     - The rationale can be identified primarily through node/edge features rather than topological structure
>
> - In such cases, our topological discrepancy measure, which excels at capturing complex structural patterns, may introduce unnecessary complexity by analyzing features (like cycle bases) that aren't relevant to the true rationale. The model ends up relying more heavily on the prediction loss other than the interpretability regularization.

---

> ### Author Response · Authors · 2024-11-25
>
> We appreciate your thorough review and hope we have adequately addressed all your concerns. If you have any additional questions or would like further clarification on any point, we welcome continued discussion.

---

> > ### Comment · Reviewer_nH1H · 2024-11-26
> >
> > Thank the authors for their detailed explanation. My concerns are mostly addressed.

---

> > > ### Author Response · Authors · 2024-11-27
> > > **Supplementary Response: Additional Experimental Results on MUTAG Dataset**
> > >
> > > Following up on our previous response regarding MUTAG performance, we conducted additional experiments that provide compelling evidence for our analysis.
> > >
> > > - Our investigation revealed that the initial lower performance on MUTAG stemmed from incorporating both 0th and 1st dimensional persistent homology features. However, the rationale subgraphs of MUTAG —- primarily NO2 and NH2 functional groups —- have relatively simple structures. Therefore, tracking higher-dimensional topological features like cycles introduced unnecessary complexity that hurt the model's performance.
> > > - We developed a simplified version of our model, TopInG-0, which uses only 0-dimensional persistent homology. The results demonstrate substantial improvements across both interpretability and accuracy metrics:
> > >
> > > Table 1: Comparison of interpretable GNN on MUTAG dataset. Best and second best ones are in bold and italic respsectively.
> > >
> > > | MUTAG    |        AUC       |        ACC       |
> > > |----------|:----------------:|:----------------:|
> > > | DIR      |   64.44 ± 28.81  |   68.72 ± 2.51   |
> > > | GSAT     |   99.38 ± 0.25   | **98.28 ± 0.78** |
> > > | GMT-LIN  | **99.87 ± 0.09** |   91.20 ± 2.75   |
> > > | TopInG   |   96.38 ± 2.56   |   92.92 ± 7.02   |
> > > | TopInG-0 |  _99.40 ± 0.07_  |  _95.18 ± 2.24_  |
> > >
> > > - As shown in the Table 1, TopInG-0 achieves the second-best performance in both interpretability (AUC) and prediction (ACC) compared to baseline interpretable GNN models. These results validate our analysis and demonstrate that our approach remains highly competitive when properly configured for molecular datasets with simpler structural patterns.
> > >
> > > We hope these supplementary experimental results better address the reviewer's concerns about our model's performance on the MUTAG dataset.  If there are any additional questions or further clarification required, we welcome continued discussion.

---

> ### Author Response · Authors · 2024-12-02
>
> Thanks again for your time and careful review. If you think we have resolved all your concerns, we will kindly ask you to consider reassessing the rating.

---

### Author Response · Authors · 2024-11-24
**General Response**

We sincerely thank all reviewers for their thorough reviews and constructive feedback. The reviewers highlighted several strengths of our work, including: (1) the innovative application of topological data analysis to interpretable GNNs (Reviewer nH1H, gpKc, rSzd, xsRV); (2) the solid mathematical foundation and theoretical guarantees for optimizing to ground truth rationale subgraphs (Reviewer nH1H, gpKc, rSzd);

There are two common concerns raised by multiple reviewers. Here we will first address them in the general response (individual reviewer-specific concerns will be addressed in separate responses later):

1. (Review nH1H, gpKc, rSzd) Clarity of the TDA preliminaries and mathematical notations: multiple reviewers noted that the TDA concepts, especially persistent homology and some mathematical notations, need clearer and more intuitive explanations.

- [Answer:] We have revised Section 2.1 to make some key concepts of TDA including graph filtrations, persistent homology, and topological feature comparison more accessible to a general audience. For interested readers, we also provide more detailed discussion in Appendix B.
We have also improved the notation clarity in Section 3 regarding topological discrepancy definition and loss functions. These revisions can be found in the pdf (highlight in blue) [(pdf link)](https://anonymous.4open.science/r/TopoEx-1EE2/OpenReview/rebuttal_update.pdf)


2. (Review nH1H, xsRV) Prior regularization design choices: Several reviewers questioned the prefixed choices and sensitivity of the hyperparameters $\mu$ in our prior regularization.

- [Answer:] Our choice of $\mu$ parameters was not empirically tuned - we set these values once and used them consistently across all experiments due to the theoretical robustness of our approach. As shown in the Table 1, varying these parameters has minimal impact on performance. The key insight is that our prior regularization acts as a clustering mechanism of edges into two clusters on [0,1], where the exact cluster centers ($\mu_1$, $\mu_2$) matter less than their separation since the edge filtration is learned and used in the topological discrepancy, thanks to the stability property of persistent homology. Any choice where $|\mu_1-\mu_2|>0$ maintains a reasonable gap should work effectively, as demonstrated by our experimental results. This differs fundamentally from previous approaches like GSAT[1] and GMT[2], offering better stability and robustness to parameter choices.

We would also like to include more example results for illustration (see [pdf link](https://anonymous.4open.science/r/TopoEx-1EE2/OpenReview/MoreInterpretationVisualization.pdf)).


Table 1: TopInG's performance on BA-2motifs with different settings of $\mu$.
| ($\mu_1$, $\mu_2$) | Interpretation (AUC) |
|-------------------|-------------------|
| (0.15, 0.65)       | 99.60 ± 0.48 |
| (0.25, 0.75)       | 100.00 ± 0.00 |
| (0.20, 0.70)       | 100.00 ± 0.00 |
| (0.30, 0.80)       | 100.00 ± 0.00 |



Reference:

[1] Miao, Siqi, et al. "Interpretable and generalizable graph learning via stochastic attention mechanism." International Conference on Machine Learning. PMLR, 2022.

[2] Chen, Yongqiang, et al. "How Interpretable Are Interpretable Graph Neural Networks?." arXiv preprint arXiv:2406.07955 (2024).

---

> ### Author Response · Authors · 2024-11-26
> **Illustrative Example Distribution of Our Learned Edge Filtration**
>
> Based on an insightful discussion with some reviewer regarding our prior regularization, we would like to share some example distributions of our edge filtration (learned on BA-HouseOrGrid-2Rnd dataset):
> - [Fig1](https://anonymous.4open.science/r/TopoEx-1EE2/OpenReview/toping_edge_singleplot.png) illustrates the distribution of learned edge filtration on a single graph.
> - [Fig2](https://anonymous.4open.science/r/TopoEx-1EE2/OpenReview/toping_edge_avgplot.png) displays an averaged distributions across 10 sample graphs
>
> Intuitively, the distribution resembles a mixture of two Gaussians as expected: a wider one on the left corresponding to $G_\epsilon$ and a narrower one on the right corresponding to $G_X$.
> We believe these illustrative examples help understand what has been learned in our model.

---

### Meta-Review · Area_Chair_3feY · 2024-12-20

**Metareview:**

This paper introduces TOPING, an interpretable GNN framework leveraging topological data analysis (TDA), particularly persistent homology, to identify stable rationale subgraphs. By employing rationale filtration learning and a novel topological discrepancy loss, TOPING separates input graphs into rationale (important) and noise subgraphs, optimizing for both prediction accuracy and interpretability. Experimental results on synthetic and real-world datasets demonstrate significant improvements over state-of-the-art methods, achieving over 20% performance gains. However, potential evaluation inconsistencies with existing methods warrant further investigation. The approach offers theoretical guarantees and addresses key limitations in current interpretable GNN models, advancing both predictive and explanatory capabilities.

Interpreting GNNs using topological data analysis is an interesting approach. However, as many reviewers pointed out, the writing can be significantly improved. Even if the negative review is discounted, the average score is 6, with all reviews being weak accepts. This indicates that the paper is a weak accept. Moreover, the paper requires significant rewriting; it is difficult to accept in its current form. Therefore, I encourage the authors to revise the paper based on the reviewers' comments and resubmit it to a reputable ML venue.

**Additional Comments On Reviewer Discussion:**

Most of the reviewers expressed concerns about the paper's writing, which the authors addressed. However, even after the discussion, the reviewers did not raise their scores. One of the negative reviewers did not respond. Even if the negative review is discounted, the average score is 6, with all reviews being weak accepts. This indicates that the paper is a weak accept. The idea of interpreting GNN with TDA is an interesting one. Thus, if there is space, it is acceptable to publish. However, the paper requires significant rewriting.

---

### Decision · Program_Chairs · 2025-01-22

Reject